# Improvement in gait stability in older adults after ten sessions of standing balance training

**Leila Alizadehsaravi**⬤¤a**, Sjoerd M. Bruijn, Wouter Muijres**¤b**, Ruud A. J. Koster, Jaap H. van Dieën**⬤*

Faculty of Behavioural and Movement Sciences, Department of Human Movement Sciences, Vrije Universiteit Amsterdam, Amsterdam, The Netherlands

¤a Current address: Faculty of Mechanical, Department of Biomechanical Engineering, Maritime and Materials Engineering, Technical University Delft, Delft, The Netherlands
¤b Current address: Human Movement Biomechanics Research Group, Katholieke Universiteit Leuven, Leuven, Belgium
* j.van.dieen@vu.nl

**Data Availability Statement:** All relevant data are within the article and its Supporting information files.

## Abstract

Balance training aims to improve balance and transfer acquired skills to real-life tasks. How older adults adapt gait to different conditions, and whether these adaptations are altered by balance training, remains unclear. We hypothesized that reorganization of modular control of muscle activity is a mechanism underlying adaptation of gait to training and environmental constraints. We investigated the transfer of standing balance training, shown to enhance unipedal balance control, to gait and adaptations in neuromuscular control of gait between normal and narrow-base walking in twenty-two older adults (72.6 ± 4.2 years). At baseline, after one, and after ten training sessions, kinematics and EMG of normal and narrow-base treadmill walking were measured. Gait parameters and temporal activation profiles of five muscle synergies were compared between time-points and gait conditions. Effects of balance training and an interaction between training and gait condition on step width were found, but not on synergies. After ten training sessions step width decreased in narrow-base walking, while step width variability decreased in both conditions. Trunk center of mass displacement and velocity, and the local divergence exponent, were lower in narrow-base compared to normal walking. Activation duration in narrow-base compared to normal walking was shorter for synergies associated with dominant leg weight acceptance and non-dominant leg stance, and longer for the synergy associated with non-dominant heel-strike. Time of peak activation associated with dominant leg stance occurred earlier in narrow-base compared to normal walking, while it was delayed in synergies associated with heel-strikes and non-dominant leg stance. The adaptations of synergies to narrow-base walking may be interpreted as related to more cautious weight transfer to the new stance leg and enhanced control over center of mass movement in the stance phase. The improvement of gait stability due to standing balance training is promising for less mobile older adults.

**Funding:** This project has received funding from the European Union's Horizon 2020 research and innovation programme under the Marie Skłodowska-Curie grant agreement No 721577. SMB was funded by a VIDI grant (016. Vidi.178.014) from the Dutch Organization for Scientific Research (NWO) and RAJK was supported by a grant of the European Research Council (grant No. 715945). The funders had no role in study design, data collection and analysis, decision to publish, or preparation of the manuscript.

**Competing interests:** The authors have declared that no competing interests exist.

# Introduction

Falls in older adults mostly occur during walking [1,2]. Thus, if effects of standing balance training programs do not transfer to improvements in gait stability, they are unlikely to decrease the number of falls. On the one hand, effects of balance training have been described as task specific, i.e., leading to performance improvement of the task that has been trained but not to improvements in other tasks [3]. On the other hand, transfer to gait stability from solely standing balance training [4–6] is suggested by improved clinical balance scores, gait parameters, and performance on the timed up and go, and other tests. In addition, studies that trained standing balance found reduced falls [7]. Consequently, the existence of transfer from standing balance training to gait, as well as the mechanisms underlying such a transfer, if present, are insufficiently clear. In addition, while improved balance has been reported even after one training session [8], the required time course for transfer is unknown.

Fall prevention training programs aim to improve balance control employing plastsicity of the neuromuscular system. To prevent falls, one needs to be able to adapt gait when facing environmental challenges, such as when forced to walk with a narrow step width. Older adults show more pronounced adaptations to narrow-base walking than young adults [11], possibly because they are more cautious in the presence of balance threats [12]. An interaction between training and stabilizing demands may be expected. On the one hand, increased confidence after training may result in less adaptation to a challenging condition. On the other hand, balance training may enhance the ability to adapt to challenging conditions. Therefore, if transfer of standing balance training to gait occurs, an altered modulation of gait between normal and narrow-base walking might be expected after training, but the direction of change is unpredictable.

Transfer of balance training to gait should become apparent in biomechanical gait parameters related to balance control. Relevant gait parameters would be variability and local dynamic stability of gait, as these were shown to be associated with a history of falls in older adults [9]. In addition, larger trunk mediolateral center of mass (CoM) excursions and velocities are expected to cause an increased fall risk [10] and both these parameters as well as their variability are larger in older than young adults [11]. Moreover, in narrow-base walking, which challenges mediolateral balance control, CoM displacement and CoM velocity are decreased and local dynamic stability is increased indicating enhanced control [11,12]. Step width and its variability reflect active balance control during gait [13]. Mediolateral gait stability is thought to be actively controlled by adjusting foot placement [14], as centre of mass kinematics during the preceding swing phase strongly correlate with foot position in the next stance phase [15,16]. The strong coupling between centre of mass kinematics and foot placement is found to decrease in conditions in which gait is stabilized, such as in external lateral stabilization, increasing confidence that lateral gait stability is indeed controlled by foot placement adjustment [17]. This decoupling coincides with a decrease in step width and step width variability [18]. Moreover, increased step width and step width variability have been found in older compared to young adults and in fallers compared to non-fallers [19–21]. Consequently, step width and step width variability are considered to be proxies of the quality of active mediolateral control of gait stability using foot placement adjustments and may therefore be useful to evaluate the effect of balance training. Transfer of balance training effects to gait could thus be reflected in increased local dynamic stability, decreased trunk CoM displacement and velocity, decreased trunk CoM displacement variability, and decreased step width and step width variability.

Motor tasks are thought to be performed through modular control of muscles, reflected in so-called muscle synergies [22]. A potential mechanism for standing balance training to affect

gait is through shared muscle synergies between standing balance and gait. Human gait has been reported to be controlled by four to eight of such muscle synergies [23–25], the combination of which shapes the overall motor output [26,27]. Standing balance is also reported to be controlled through a limited number of muscle synergies [25], and it has shared synergies with gait [25]. These shared synergies were identified as subserving postural ability. Hence, standing balance training could change synergies involved in standing balance and, by extension, the shared synergies involved in gait. The overlap in modular control provides a mechanism through which standing balance training can influence gait stability.

Muscle synergies consist of time-dependent patterns (activation profiles) and time-independent components (muscle weightings). Motor adaptation is assumed to result from altering either of these in response to task and environmental demands [28,29]. Furthermore, due to aging and related changes in sensory and motor organs, adapted synergies are likely required to maintain motor performance [30,31]. Synergy analysis of gait and gait related-tasks revealed a less efficient modular control in older compared to young adults [30–32]. To adapt to environmental challenges, ~~similarly~~ widened activation profiles appear to be used [29,32], indicating a more robust control and suggesting that the age effects described reflect a more cautious control. Balance training might alter synergies in gait and the adaptation of these synergies to task demands, as has been shown in a comparison between expert and novice dancers [33,34].

Previously, we have shown that training of standing balance control improved balance robustness. This was defined as the time to balance loss in unipedal standing on a platform with decreasing rotational stiffness around a sagittal axis. Balance robustness increased by 33% already after one training session, with no further improvement after ten sessions. Balance performance, defined as absolute mediolateral center of mass velocity, was improved by 19% in perturbed unipedal standing after one session and by 18% in unperturbed unipedal standing after ten sessions [35]. In the current study, we aimed to investigate the transfer of effects of standing balance training to normal and narrow-base walking in older adults, as well as the adaptation of older adults to narrow-base walking. The modulation of balance control between two conditions aimed to test adaptability of balance control to environmental constraints and effects of training were studied to analyse the plasticity of balance control. To this end, we evaluated normal walking and narrow-base walking on a virtual beam, both on a treadmill, before training and after one and ten training sessions. We focused on mediolateral balance control, because larger mediolateral instability has been shown to be associated with falls in older adults [36,37], narrow-base walking challenges mediolateral stability and also standing balance was evaluated in the mediolateral direction. To assess performance in narrow-base walking, we calculated foot placement errors, the percentage of steps in the beam, step width, and the step width variability [38]. For both gait conditions, we calculated trunk CoM displacement and trunk CoM displacement variability, trunk CoM velocity, and the local divergence exponent of trunk movement as measures of gait stability.

We extracted muscle synergies to characterize effects of training on the neuromuscular control of gait and on adaptations to narrow-base walking. We focused on changes in timing of muscle activation by assessing the synergy activation profiles' full width at half maximum (FWHM), reflecting activation duration, and Center of Activation (CoA), reflecting time of peak activation. We hypothesized that adaptations to narrow-base walking would be reflected in enhanced gait stability [39] and in more prolonged muscle activation. Furthermore, we hypothesized that training effects would transfer to gait as reflected in improved gait stability and narrow-base walking performance and activation profiles with less prolonged muscle activation. These effects were expected to be more pronounced in narrow-base walking compared to normal walking and more pronounced after ten training sessions compared to a single session.

We would like to emphasize that the outcome variables step width and step width variability were added to the analysis based on reviewer feedback. In contrast to other variables, these showed a significant effect of training. Even though step width and step width variability seem to be sensitive to quality of balance control in gait, step width measures are not necessarily affected by postural balance training. Postural balance is controlled using torques around the ankle and hip of the standing leg [40,41], but does not include stepping (i.e. foot placement). However, as in postural balance, the stance leg is used to regulate stabilizing torques in the stance phase gait [42]. This stance leg control co-determines the control of foot placement [43] and therefore reduced step width and step width variability are likely results of improved control by the stance leg. These predictions on step width and step width variability were conceived, based on reviewer comments, after the data were processed and analyzed. These predictions are thus explorative and aim to drive future research [44,45] and we ask the reader to consider the limitations of the evidence provided by these variables. While this result is in line with our primary hypothesis and indicates transfer of effects of standing balance training to gait, in a strict sense it cannot be considered a planned analysis.

## Methods

The methods described here partially overlap with those described in a previous paper [35], as data were obtained in the same cohort.

### Participants

Twenty-two older (72.6 ± 4.2 years old; mean ± SD, 11 females) healthy volunteers, without a history of falls in the preceding year, participated in this study. The required sample size was estimated at twenty-two based on power analysis for an F test of a repeated measures ANOVA, assuming a Cohen's f of 0.44 (based on meta-analysis of the effect of standing balance training on steady-state balance [46]) and a correlation among repeated measures of 0.6 ($\beta = 0.8$, G $^*$ power 3.1.9.2, Düsseldorf, Germany), comparable to similar studies [47,48]. Participants were recruited through a radio announcement, by contacting older adults who had previously participated in our research, and via flyers and information meetings. Individuals with obesity (BMI > 30), cognitive impairment (MMSE < 24), peripheral neuropathy, a history of neurological or orthopaedic impairment, use of medication that may negatively affect balance, inability to walk for 3 minutes without aid, and performing sports with balance training as an explicit component (e.g., Yoga or Pilates) were excluded. All participants provided written informed consent before participation and the procedures were approved by the ethical review board of the Faculty of Behavioural & Movement Sciences, VU Amsterdam (VCWE-2018-171).

### Experimental procedures

Participants completed an initial measurement to determine baseline values (Pre), and a single-session individual balance training (30-minutes), a second measurement (Post1) to compare to baseline to assess single-session training effects. The program was continued with a 3-week balance training (9 sessions x 45 minutes training), and a third measurement (Post2) to compare to baseline to assess 10-session training effects (Fig 1).

The measurements consisted of one experimental condition on a robot-controlled platform (standing balance) and two experimental conditions performed on a treadmill: virtual-beam walking (Fig 2) and normal walking.

The training sessions consisted of exercises solely focused on unipedal balancing with blocks of 40–60 seconds exercises in which balance was challenged by different surface

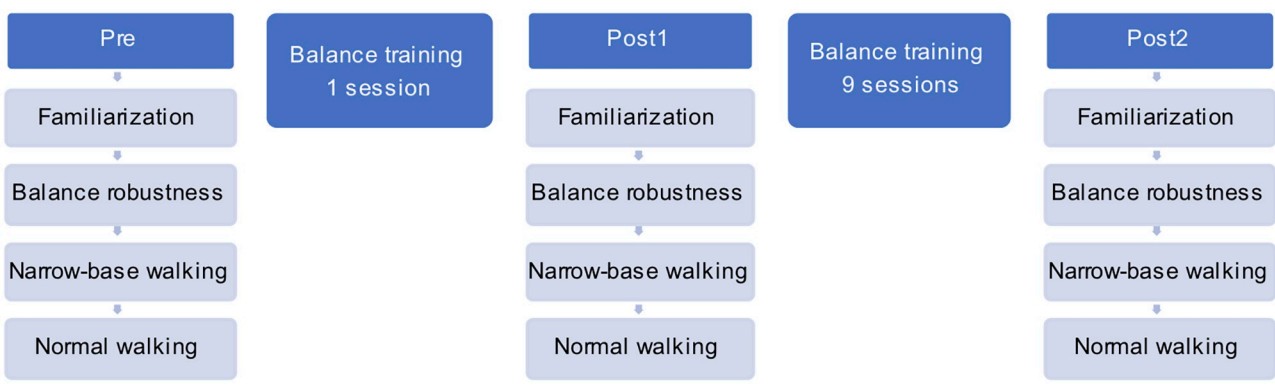

**Fig 1. Block diagram of the study; training and gait assessment.**

conditions, static vs dynamic conditions, self-perturbations and external perturbations while catching a ball in a dual tasking exercise (e.g. catching, throwing and passing a ball) [49]. Participants performed the exercises in a group (except for the first, individual session) and were always under supervision of the physiotherapist in our research team (for details see S1 File).

## Instrumentation and data acquisition

Balance robustness and performance were evaluated using a custom-made balance platform controlled by a robot arm (HapticMaster, Motek, Amsterdam, the Netherlands) and results were reported previously [35]. To quantify transfer to gait, participants were instructed to walk for 4.5 minutes on a treadmill with an embedded force plate. For estimating the local divergence exponent, a minimum of 150 steps is recommended [50]. We expected that a total duration of 4.5 minutes was needed to reach that number. To avoid effects of gait speed on outcome measures, this was kept constant at 3.5 km/h for all participants [51].

For safety reasons, handrails were installed on either side of the treadmill, and an emergency stop button was placed within easy reach (MotekForcelink, Amsterdam, the Netherlands). We assessed walking in two conditions, normal walking and narrow-base walking, in a randomized order, with a minimum of two minutes seated rest in between conditions. A narrow-base walking paradigm was chosen because narrow-base walking has been shown to challenge mediolateral stability in older adults [12,34]. In this condition, participants were instructed to place their entire foot inside a green light-beam path (12 cm width) projected in the middle of the treadmill (Bonte Technology/ForceLink, Culemborg, The Netherlands) as accurately as possible [38]. Participants were acquainted with the experimental setup to minimize habituation effects. For familiarization, participants performed 30 seconds of narrow-base walking before the measurement.

Kinematic data were obtained by two Optotrak 3020 camera arrays sampling at 50 Hz (Northern Digital, Waterloo, Canada). Ten active marker clusters (3 markers each) were placed on the posterior surface of the thorax (1), pelvis (1), arms (2), calves (4), and feet (2) (Fig 2). Positions of anatomical landmarks were digitized by a 4-marker probe and a full-body 3D-kinematics model of the participant was formed relating clusters to the anatomical landmarks [52]. The position of the foot segments was obtained through cluster markers on both feet, digitizing the medial and lateral aspects of the calcaneus, and the heads of metatarsals one and five [38]. Additionally, to allow assessment of performance in narrow-base walking, position and orientation of the projected beam were determined by digitizing the four outer bounds of the beam on the treadmill.

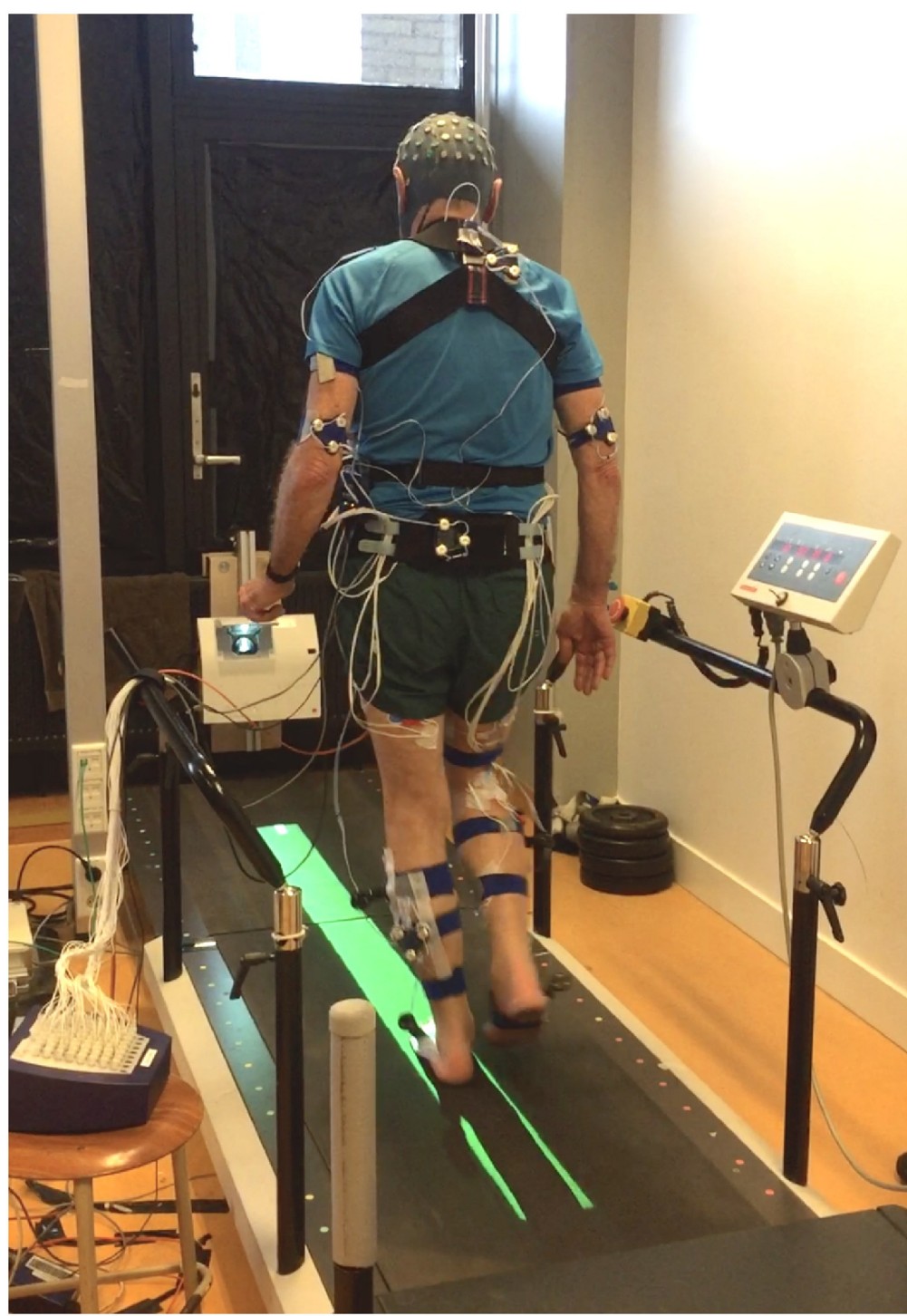

**Fig 2. Narrow-base walking on a treadmill.**

Surface electromyography (EMG) data were recorded from 11 muscles; 5 unilateral muscles of the dominant leg: tibialis anterior (TAD), vastus lateralis (VLD), lateral gastrocnemius (GLD), soleus (SOD), peroneus longus (PLD), and 6 bilateral muscles: rectus femoris (RFD, RFN), biceps femoris (BFD, BFN), and gluteus medius (GMD, GMN). These muscles were

selected based on a previous study that showed changes in walking synergies due to long-term training [34]. The ankle muscles were chosen for their key role in postural stability [53–55], and the gluteus medius for its role in mediolateral stability [56].

Bipolar electrodes were placed in accordance with SENIAM recommendations [57]. EMG data were sampled at a rate of 2000 Hz and amplified using a 16-channel TMSi Porti system (TMSi, Twente, The Netherlands). The dominant leg was the preferred stance leg for unipedal stance. The preferred stance leg was reported by the participant prior to the experiment and confirmed by the experimenter by asking the participant to kick an imaginary soccer ball. The supporting leg was considered the preferred stance leg [35]. Focus was on this leg, because we extensively assessed unipedal balance control on this leg as reported earlier [35].

## Data analysis

**Gait events.** The first 30 seconds of all gait trials were removed, to avoid habituation effects. Heel-strikes were detected through a peak detection algorithm based on the center of pressure [58]. This algorithm proved to be precise when the center of pressure moved in a butterfly pattern. However, for narrow-base walking, the feet share a common area in the middle of the treadmill. Therefore, identifying which leg touched the surface was problematic, as the butterfly pattern was not formed in narrow-base walking. Hence, heel-strikes were detected based on the center of pressure peak detection, but the associated leg was identified based on kinematic data of the foot marker. 160 strides per participant per condition were used to calculate all gait variables (i.e., stability variables and muscle synergies).

**Gait stability.** To quantify gait performance, we evaluated foot placement, step width, and CoM behaviour. Evaluation of foot placement was only performed for narrow-base walking. We assessed foot placement error, determined as the mean mediolateral distance of the furthest edge of the foot from the edge of the beam. If the entire foot was within the beam the error was set to zero. The foot placement error was used because it is by design a direct quantification of task performance. Hence, we expected it to be the most sensitive measure to changes in task performance. In addition, the percentage of steps inside the beam was defined as the number of the steps in which the whole foot was placed within the beam. To assess foot placement in both walking conditions, step width and its variability were determined. These measures were computed as the mean and standard deviation of the distance between the mediolateral position of the left and right foot over 160 strides. Additionally, the trajectory of the CoM of the trunk was estimated from the mediolateral movement of the trunk markers [15,59]. From this, we calculated mean and standard deviation of the peak-to-peak mediolateral trunk CoM displacement and mean of CoM velocity per stride. Trunk CoM displacement, displacement variability, and velocity are commonly used measures to express postural stability. However, particularly for use during gait these measures are not entirely undisputed as they are not necessarily minimized for task execution. Hence, local dynamic stability was also evaluated using the LDE, as evidence suggests its validity in the context of gait stability and falling [50]. Calculation of the LDE was based on Rosenstein's algorithm [60,61]. We used the time normalized time-series (i.e., 160 strides of data were time normalized to 16000 samples, preserving between stride variability) of trunk CoM velocity to reconstruct a state space with 5 embedding dimensions at 10 samples time delay [59]. The divergence for each point and its nearest neighbour was calculated and the LDE was determined by a linear fit over half a stride to the averaged log transformed divergence.

**Muscle synergies.** EMG data were high-pass (50 Hz, bidirectional, 4th order Butterworth) [29] and notch filtered (50 Hz and its harmonics up to the Nyquist frequency, 1 Hz bandwidth, bidirectional, 1st order Butterworth). The filtered data were Hilbert transformed, rectified and

low-pass filtered (20 Hz, bidirectional, 2nd order Butterworth). Each channel was normalized to the maximum activation obtained for an individual per measurement point per trial. Synergies were extracted from 11 muscles using non-negative matrix factorization based on Lee and Seung's multiplicative update rule [62] with 50 repetitions with a maximum of 1000 iterations to update the components and at a tolerance of $10^{-6}$. Five synergies were extracted from the whole dataset to account for a minimum of 85% of the variance in the EMG data (Fig 7). It has been shown that perturbations during walking change the temporal activation profiles as compared to normal walking, while muscle weightings are preserved [63]. Therefore, in the current study we fixed muscle weightings between conditions and time-points to be able to identify changes in the temporal activation. These muscle weightings were extracted from the concatenated EMG data of both conditions at all time-points. This allowed for objective comparison of synergy activation profiles between normal and narrow-base walking and between time-points. The time-normalized EMG data of the muscles $E_{11 \times (3 \times 2 \times 100 \times 160)}$, was factorized to two matrices: time-invariant muscle weightings, $W_{11 \times 5}$, and temporal activation profiles of the factorization, $A_{5 \times (3 \times 2 \times 100 \times 160)}$, where 11 was the number of muscles, 3 the number of time-points, 2 the number of conditions, 100 the number of samples in each stride,160 the number of strides, and 5 the number of synergies.

To compare activation profiles, we evaluated the FWHM per stride for each activation profile. The FWHM is defined as the number of data points above half of the maximum of the activation profile, after subtracting the minimum activation [64]. In addition, we evaluated the CoA (indicating the center of the distribution of activation timing within a gait cycle) per stride, defined as the angle of the vector that points to the center of mass in the activation profile transformed to polar coordinates [65]. The FWHM metric reflects the duration of activation but is naïve of timing of activation. The CoA metric reflects the timing of activation but is naïve of duration of activation. FWHM and CoA were averaged over 160 strides per participant per condition. For CoA data, circular averaging was used.

## Statistics

Shapiro-Wilk's test was performed on all measures. In case of non-normally distributed data, a log transformation was performed. One-way repeated measures ANOVA were performed to investigate the effects of Training (Pre, Post1, Post2) on foot placement errors, and % of steps within the beam. Post hoc comparisons (paired sample t-tests), with Holm's correction for multiple comparisons were performed to investigate the effect of one and ten training sessions (Pre vs Post1 and Pre vs Post2, respectively).

Two-way repeated-measures ANOVAs were used to identify effects of Training (Pre, Post1, Post2) and Condition (normal and narrow-base walking), as well as their interaction on step width; step width variability; FWHM; and on trunk kinematics: CoM displacement, CoM displacement variability, CoM velocity, and LDE. When the assumption of sphericity was violated, the Greenhouse-Geisser correction was used. In case of a significant effect of Training, or an interaction of Training x Condition, post hoc tests with Holm's correction for multiple comparisons were performed. To identify effects of Training and Condition (normal and narrow-base walking), as well as their interaction, on CoA a parametric two-way ANOVA for circular data was used using the Circular Statistic MATLAB toolbox [66]. In all statistical analyses $\alpha = 0.05$ was used.

## Results

One participant was not able to perform the treadmill walking trials for the full duration and data for this participant were excluded.

## Gait performance

Performance in narrow-base walking, as reflected in foot placement errors, did not change significantly with Training ($F_{2,40}$ = 1.479, p = 0.242; Fig 3a). Also, the percentage of the steps within the beam did not change significantly with Training ($F_{2,40}$ = 2.934, p = 0.065; Fig 3b). However, Training had a significant effect on step width ($F_{1.57,31.36}$ = 7.121, p = 0.005; Fig 4a). Moreover, the interaction of Training and Condition significantly affected step width ($F_{1,20}$ = 261.075, p < 0.001). Post hoc analyses showed that step width decreased after ten sessions and only in narrow-base walking (t = 4.062, p < 0.001), and step width was smaller in narrow-base compared to normal walking at all time-points (t = -12.302, p < 0.001; t = -11.763, p < 0.001; t = -14.386, p < 0.001; for Pre, Post1, and Post2, respectively).Training also had a significant effect on step width variability ($F_{2,40}$ = 8.724, p < 0.001; Fig 4b). Post hoc testing showed that step width variability had not significantly changed after one session (t = 0.898, p = 0.375), but was decreased after ten sessions (t = 3.982, p < 0.001). There was no significant effect of Condition or interaction of Training and Condition on step width variability ($F_{1,20}$ = 0.576, p = 0.457; $F_{2,40}$ = 1.994, p = 0.149).

Training did not significantly affect trunk CoM displacement, displacement variability, and velocity ($F_{2,40}$ = 2.729, p = 0.082; $F_{2,40}$ = 0.469, p = 0.628; $F_{2,40}$ = 2.024, p = 0.145). Condition significantly affected all three variables, with lower displacement and velocity ($F_{1,20}$ = 96.007, p < 0.001; $F_{1,20}$ = 168.26, p < 0.001; respectively, Fig 5a & 5b), but larger CoM displacement variability ($F_{1,20}$ = 4.678, p = 0.042, Fig 5c), in narrow-base compared to normal walking. No significant interactions of Training x Condition were found (p > 0.05). Training did not significantly affect LDE ($F_{2,40}$ = 0.205, p = 0.814), but Condition did, with lower values (indicating improved stability) in narrow-base compared to normal walking ($F_{1,20}$ = 26.223, p < 0.001; Fig 6). No significant interaction of Training x Condition was found for the LDE ($F_{1.3,24.699}$ = 3.112, p = 0.078).

## Muscle synergies

Five muscle synergies were extracted with a fixed muscle weighting matrix **W** (Fig 7) and activation profiles per individual per condition and time-point (Fig 8). This accounted for 87 ± 2% of the variance in the EMG data. Based on muscle weightings and activation profiles, the first synergy was predominantly active in the stance phase of the dominant leg, with major involvement of soleus and gastrocnemius lateralis. The second synergy was active during the weight acceptance phase of the dominant leg, where the quadriceps (vastus lateralis, rectus femoris) muscles were most engaged. The third synergy resembled partial mirror images of synergies 1 and 2 for the non-dominant leg but differed, because only a subset of muscles was measured. It was mainly active in the stance phase of the non-dominant leg, with major involvement of the gluteus medius and rectus femoris. It lacks muscle activation during the push-off (represented in synergy 1), because lower leg muscles were not measured and represented thigh muscle activity related to weight acceptance (represented in synergy 2). The fourth synergy was activated prior to dominant leg heel-strike with engagement mostly of the biceps femoris of the dominant leg. Finally, the fifth synergy appeared to be the mirror image of the fourth synergy, with pronounced engagement of the biceps femoris of the non-dominant leg.

**FWHM.** None of the FWHMs were significantly affected by Training. FWHMs were found to be smaller in narrow-base compared to normal walking in the synergies associated with weight acceptance of the dominant leg (synergy 2, $F_{1,20}$ = 92.86, p < 0.001) and the stance phase of the non-dominant leg (synergy 3, $F_{1,20}$ = 17.06, p < 0.001; Fig 9). In contrast, FWHM of synergies associated with heel-strike in narrow-base compared to normal walking was only

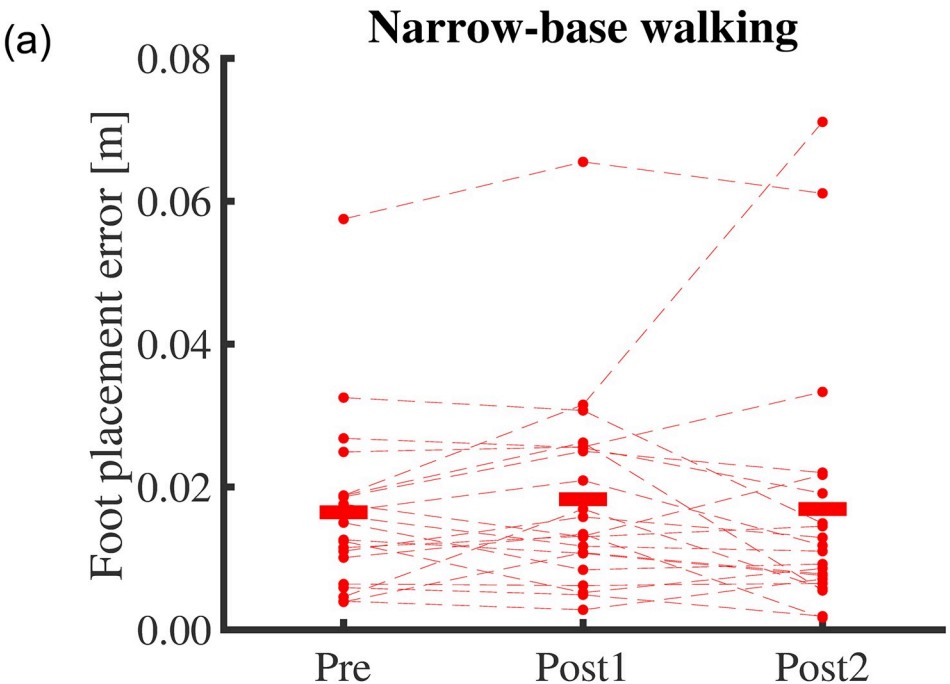

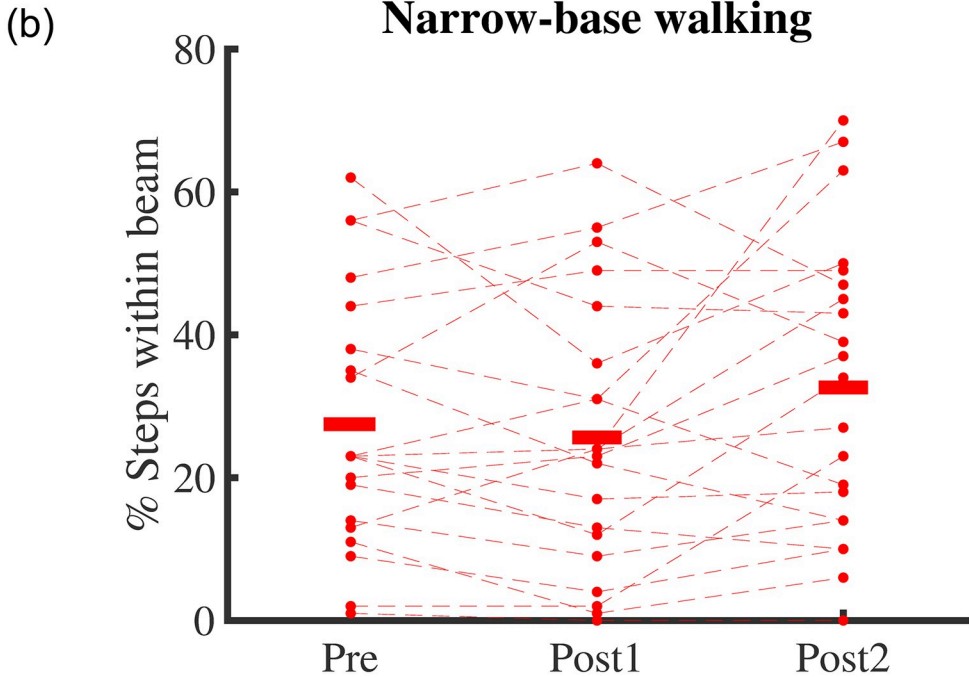

**Fig 3.** (a) Foot placement error and (b) percentage of the steps within the beam in narrow-base walking at time-points Pre, Post1, and Post2. Thin lines represent individual subject data. Red horizontal lines indicate means over subjects.

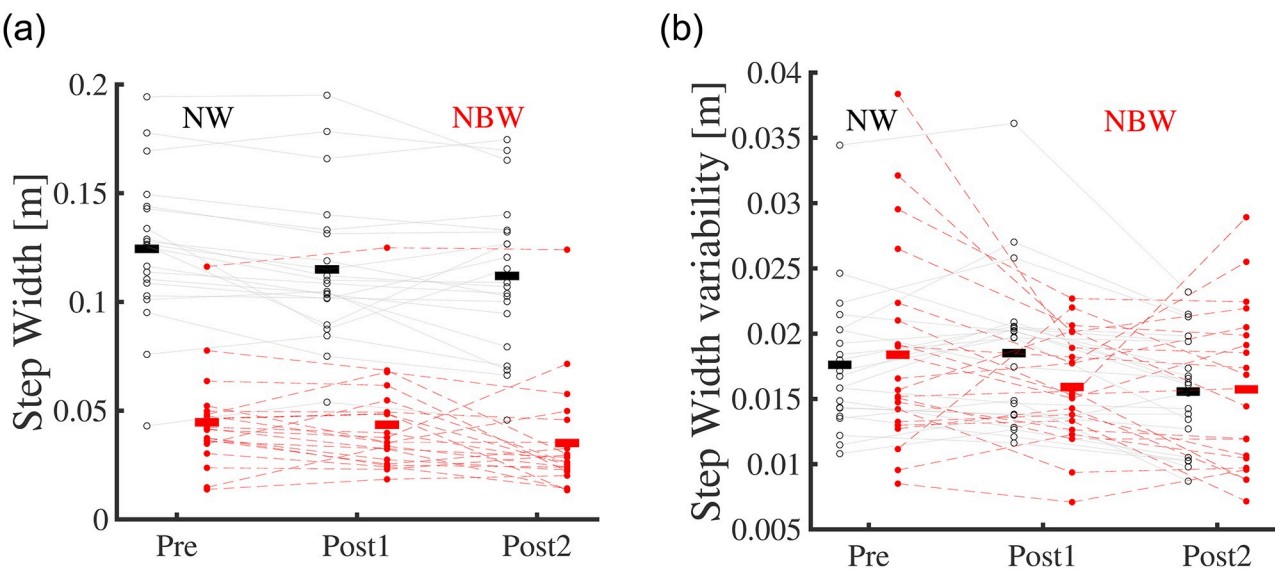

**Fig 4.** (a) Step width (b) Step width variability in narrow-base and normal walking at time-points Pre, Post1, and Post2. Thin lines represent individual subject data. Thick horizontal lines indicate means over subjects. Black, normal walking; red filled-dot, narrow-base walking.

greater for the non-dominant leg (synergy 5, $F_{1,20} = 8.603$, $p = 0.008$) and not the dominant leg (synergy 4, $F_{1,20} = 2.198$, $p = 0.153$; Fig 9). In none of the synergies, FWHM was significantly affected by the interaction of Training x Condition ($p > 0.05$).

**CoA.** None of the CoAs were significantly affected by Training ($p > 0.05$). CoA of synergy 1, associated with dominant leg stance, occurred significantly earlier in narrow-base compared to normal walking ($F_{1,20} = 6.005$, $p = 0.015$; Fig 9). CoAs of synergy 3 ($F_{1,20} = 9.832$, $p = 0.002$), associated with non-dominant stance leg, and synergies 4 ($F_{1,20} = 22.109$, $p < 0.001$) and 5 ($F_{1,20} = 18.308$, $p < 0.001$), associated with heel-strike, were delayed in narrow-base compared to normal walking (Fig 9).

## Discussion

We studied whether effects of standing balance training transferred to gait in older adults. Additionally, we investigated adaptations in neuromuscular control of gait in older adults between normal and narrow-base walking, and the effect of one and ten sessions of standing balance training on this. We expected the neural mechanisms underlying balance in standing and walking to be adaptable (modulated between conditions) and plastic (modified by training). We also expected transfer of training effects to be most pronounced in the narrow-base condition, given its challenging nature, and higher resemblance to unipedal standing balance.

We have previously found improvements in robustness of standing balance already after one session and improvements in performance of standing balance after the first session and after ten sessions of standing balance training [35]. Here, we found a decreased step width in narrow-base walking and decreased step width variability in narrow-base and normal walking after ten sessions of training. However, we found no changes in gait synergies after training. In line with our expectations, we found increased activation duration in synergies during narrow-base walking, which suggests enhanced stability in narrow-base walking compared to normal walking.

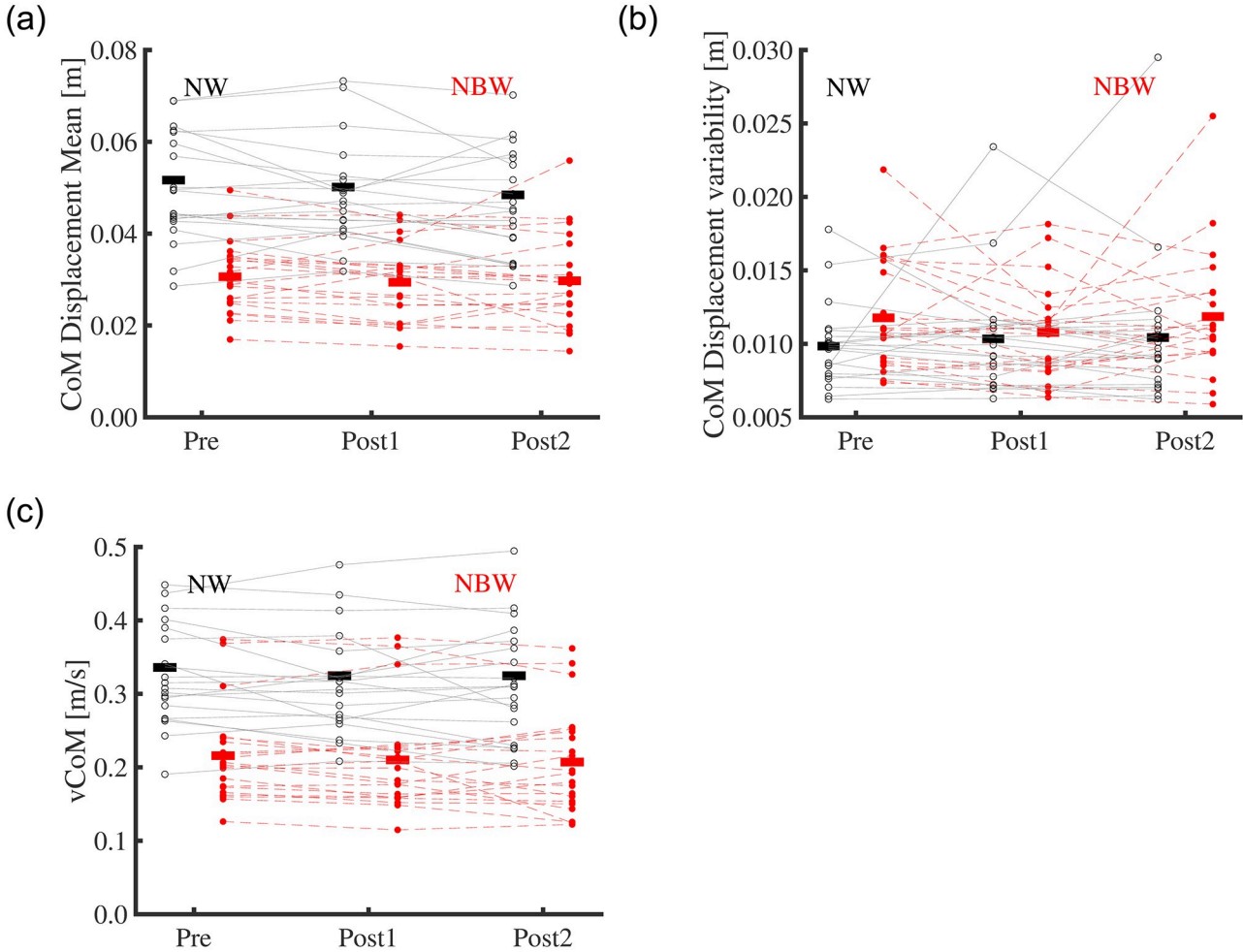

**Fig 5.** (a) Mediolateral center of mass displacement and (b) variability, and (c) center of mass velocity in narrow-base and normal walking at time-points Pre, Post1, and Post2. Thin lines represent individual subject data. Thick horizontal lines indicate means over subjects. Black, normal walking; red filled-dot, narrow-base walking.

### Transfer of training effects

As previously reported, we found improved balance robustness and performance after standing balance training [35]. In this study, after ten sessions we found transfer from standing balance training to gait. This transfer effect was manifested in a decreased step width in narrow-base walking and a decreased step width variability in both gait conditions.

Our results showed that although the participants did not better comply with the task instruction (to step within the beam), they did use narrower steps in line with those instructions. Foot placement errors did not show improvement, despite an ample scope for improvement. When interpreting the 1.5 cm mean step error, one should consider that the width of the beam was 12 cm, while the average foot width is about 10 cm [67]. This means that participants, on average, had a 2 cm margin to achieve a zero error in foot placement. Additionally, the percentages of steps inside the beam were 27.6% (SD 17.8%) pre-training and 33.3% (SD 20.1%) post-training. Although these differences were not significant (p = 0.11), the numbers show an increasing trend and suggest that there is no ceiling effect. Also, the best-recorded performance was 67% of steps within the beam, so scores much higher than the mean were

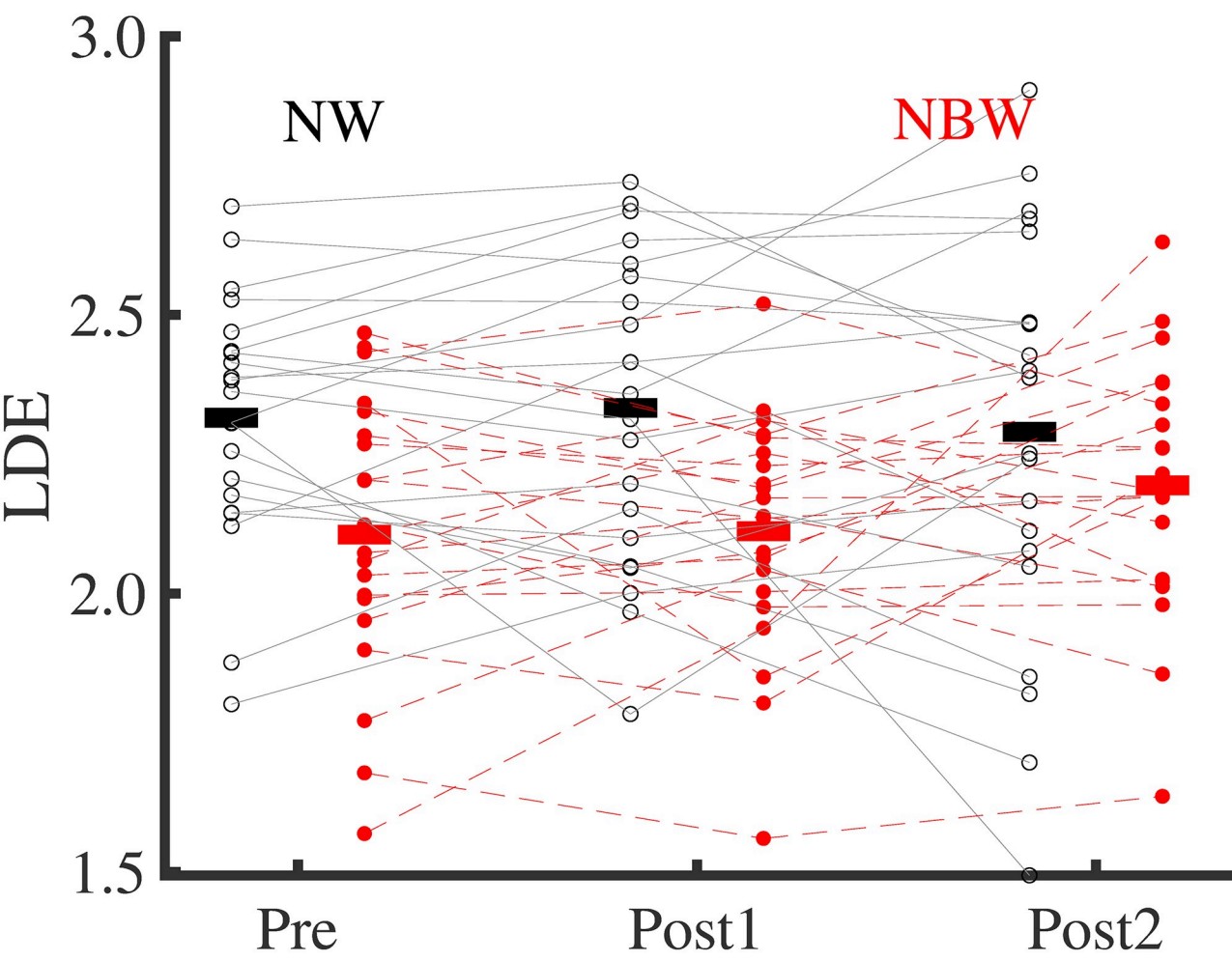

**Fig 6. Local divergence exponents in narrow-base and normal walking at time-points Pre, Post1, and Post2.** Thin lines represent individual subject data. Thick horizontal lines indicate means over subjects. Black, normal walking; red filled-dot, narrow-base walking.

achievable. The significant effect of training on step width during narrow-base walking indicates that performance improved in line with the instructions, but the improvement was not large enough to be detected by foot placement error and steps outside the beam, probably as both variables discard information on foot placement within the beam.

A reduced step width was observed only in narrow-base and not in normal walking, which might suggest that the transfer occurs more readily to a task similar to the trained task. The challenge in narrow-base walking is in nature similar to unipedal standing, since foot placement control for gait stability [14] is constrained and the CoM needs to be controlled relative to a narrow base of support during the single support phase. Nevertheless, normal walking also involves some degree of control of center of mass movement during single support [43], and a decreased step width variability in both conditions indicates transfer to this task as well. Foot placement variability has been associated with fall risk [68–70]. So, this decrease can be interpreted as a positive training outcome.

Other kinematic gait parameters were not affected by training. This might suggest that these parameters are insensitive to training. However, they were sensitive to differences between narrow-base and normal gait conditions. Previously, improved gait parameters were

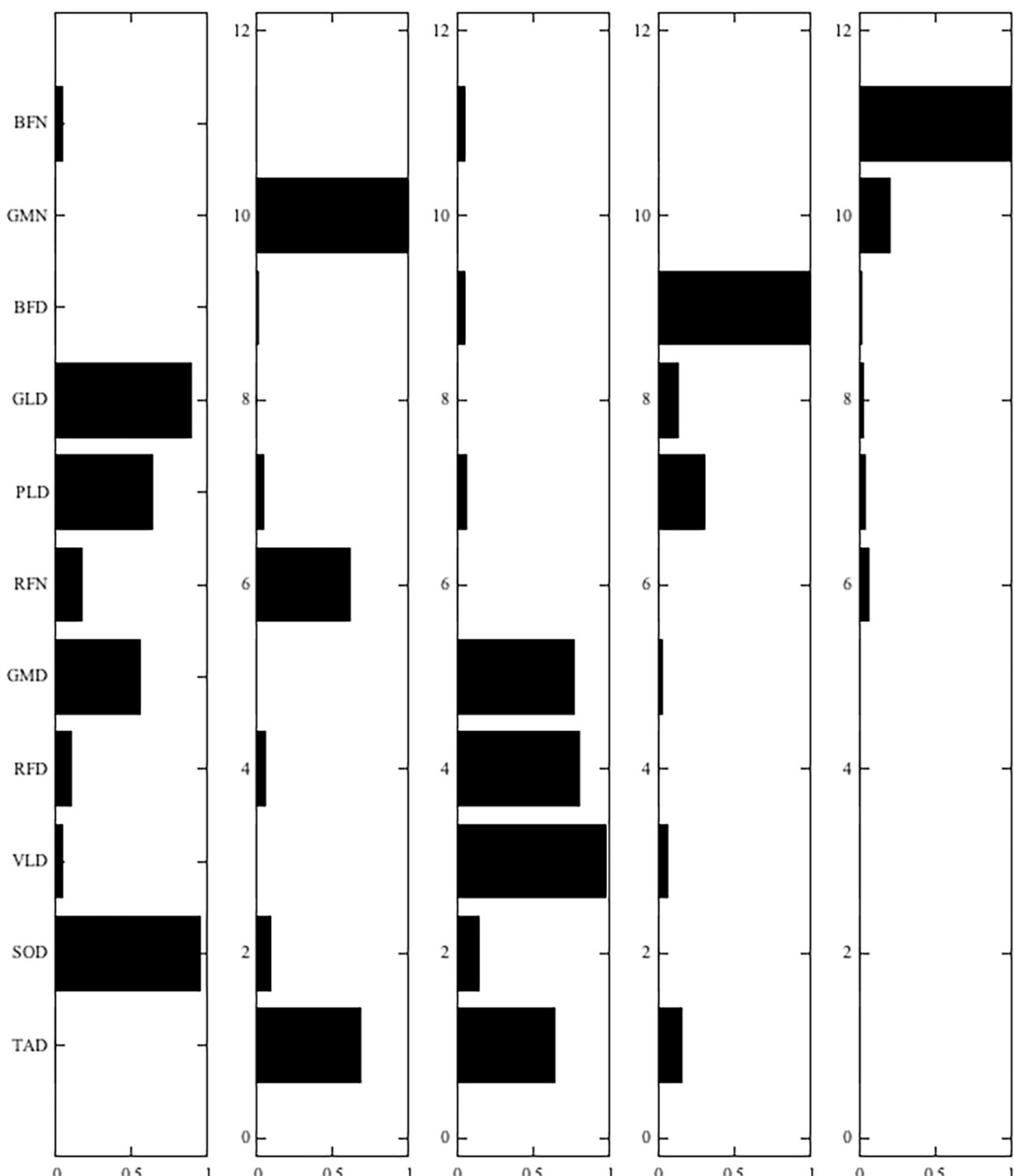

**Fig 7. Time-invariant muscle weightings of synergies extracted from concatenated data, over all individuals, conditions, and time-points.**
Muscles monitored unilaterally on the dominant side (D): tibialis anterior (TA), vastus lateralis (VL), lateral gastrocnemius (GLD, soleus (SO), and peroneus longus (PLD). Muscle collected on the dominant (D) and non-dominant side (N): rectus femoris (RFD, RFN), biceps femoris (BFD, BFN), and gluteus medius (GMD, GMN).

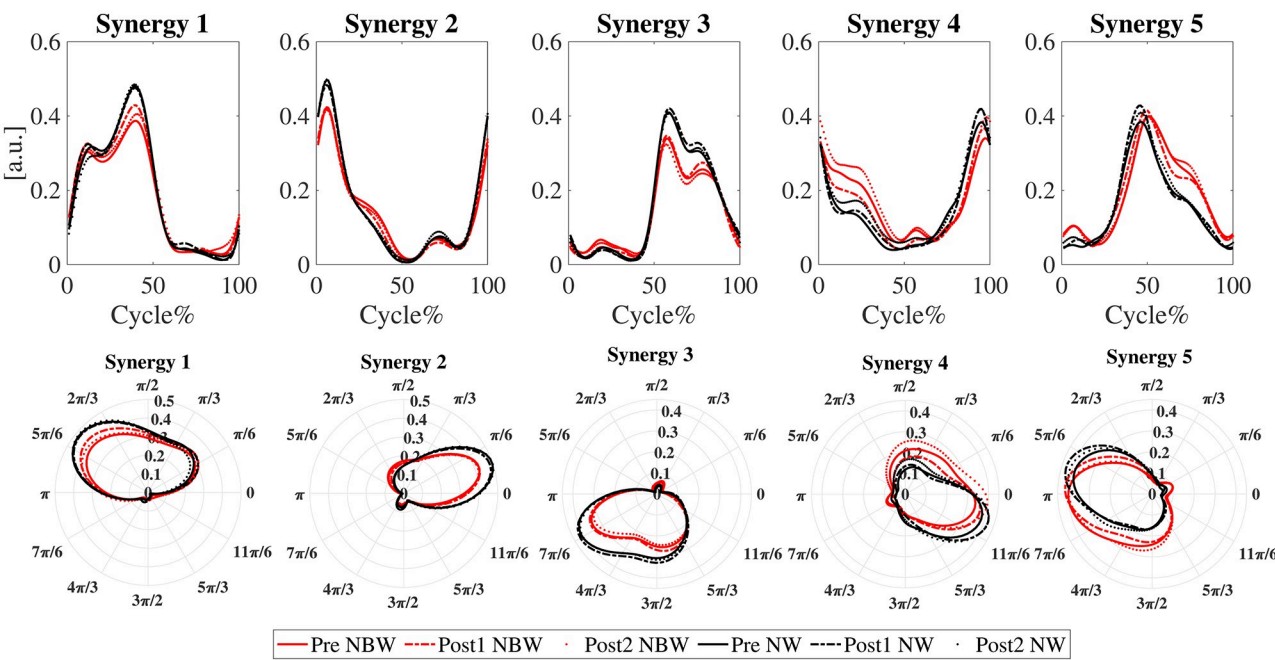

**Fig 8. Activation profiles of the extracted synergies as time series and in polar coordinates in narrow-base and normal walking at time-points Pre (solid), Post1 (dash-dot), and Post2 (dotted).** The x-axis in the Cartesian coordinates represents one gait cycle. One gait cycle in polar coordinate is [0, 2π]. Black, normal walking; red, narrow-base walking.

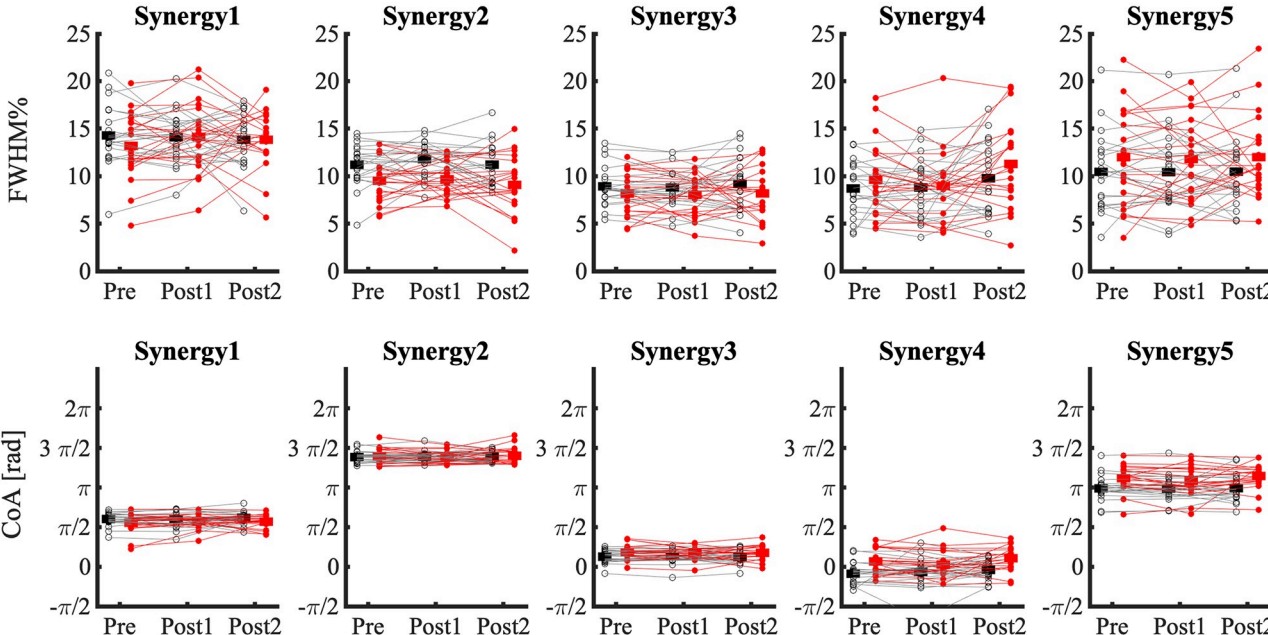

**Fig 9. FWHM and CoA of five synergies, in narrow-base and normal walking at time-points Pre, Post1, and Post2.** Thin lines represent individual subject data. Thick horizontal lines indicate means over subjects. Black, normal walking; red, narrow-base walking.

reported after 5 to 12 weeks of balance training [4,71]. Therefore, a longer duration of training might have led to more improvements in mediolateral gait stability, but more studies are needed to confirm this speculation.

Even though we found evidence of transfer, gait synergies were not affected by training. Synergy metrics may be insensitive, because subtle variations in the activation of some muscles, for example through improved feedback control, underlie improved balance. Alternatively, changes may have occurred in muscles that we did not include in our measurements. Young adults with years of experience in balance training in ballet showed different motor modules during narrow-base walking compared to novice ballet dancers [25], indicating that long-term effects of training may be apparent in muscle synergies. Note, however, that these ballet dancers' training experience most certainly included gait-related tasks [25]. In addition, the dancers had trained for at least ten years.

The lack of changes in synergies, despite modulated synergies between narrow-base and normal walking, may suggest that FWHM and COA are not sensitive to differences that occur with training. This could be due to several factors. One potential factor could be the decreased robustness of the synergies analysis when EMG data collection is performed on different days. This could hamper comparison between time-points, especially if EMG is not normalized to maximum voluntary activation. We chose not to measure the maximum voluntary activation of 11 muscles, to avoid that muscle fatigue would cause inadequate balance control during the main experiment. Given that participants were healthy older adults and the training comprised of only ten sessions spread over three weeks, we did not expect any changes in the number of synergies pre- and post-training. Changes in the number of synergies are observed typically only in individuals with severe movement disorders [72]. Further investigation is required to identify the underlying mechanisms of inter-task transfer.

The training program used excluded all exercises that directly targeted gait stability, solely focusing on the transfer of balance skill to gait as a result of standing balance training. It used unstable surfaces to make sure that the training was challenging and challenges were incremented according to guidelines in literature [73]. The current indication of transfer effects offers a valuable clue for clinicians and future studies, as exercises of standing balance form a substantial component of many training programs used in practice and showed potential to be used in training of older adults with limited mobility (to start with stationary exercises and transfer the training effects to walking). Our results highlight the necessity to optimize training methods and duration. More significant improvements in balance skills may be required to transfer acquired skills to daily-life tasks.

## Adaptation to narrow-base walking

We expected that neuromuscular control in older adults would be sufficiently plastic to adapt to narrow-base walking. In line with literature [11], our participants appeared to control CoM movements more tightly during narrow-base walking than during normal walking, as reflected in a lower CoM displacement and velocity, lower step width, and higher local dynamic stability (i.e. lower LDE). Furthermore, again in line with literature [11], variability of CoM displacement was larger in narrow-base walking. This larger variability might reflect on-line corrections of the CoM trajectory to match it to the constrained foot placement. Confronted with a narrower base, older adults reduced mediolateral CoM displacement and velocity more than young adults [11]. The LDE was also lower in narrow-base walking compared to normal, implying higher local dynamic stability, or in other words a faster attenuation of perturbation effects. This effect has previously been reported for young adults [74] but apparently older adults manage to achieve a qualitatively similar adaptation. Overall, the decreased center of

mass displacement and velocity and the faster attenuation of perturbation effects, reflected in the higher LDE, would facilitate dealing with the challenge of this condition.

The mechanical changes observed in narrow-base walking were accompanied by changes in the neuromuscular control of gait. An increase of the FWHM has been suggested to increase the ability to deal with mechanical perturbations of the gait pattern [29]. We did find differences in the FWHM of the activation profiles between narrow-base walking and normal walking. However, during narrow-base walking our participants only increased the FWHM of the activation profile associated with the non-dominant leg heel-strike (synergy 5), although a similar tendency could be observed for the dominant leg (synergy 4). These adaptations of the activation profiles may reflect enhanced control over foot placement or preparation for weight acceptance on the new stance leg in the narrow-base condition. In contrast, participants shortened the FWHM of the activation profiles associated with the stance phase of the non-dominant leg and weight acceptance of the dominant leg. These synergies share muscle activation related to weight acceptance and the change in the activation profiles is mainly visible in a slower build-up of muscle activity (Fig 8). This may reflect a slower weight acceptance by the new support leg, possibly related to the lower activation peak during push-off observable in synergy 1.

Also, the CoA of the activation profiles was different between normal and narrow-base walking. Narrow-base walking coincided with an earlier CoA of the activation profile associated with dominant leg stance (synergy 1) and delayed CoAs of the activation profile associated with dominant and non-dominant leg heel-strikes (synergies 4 and 5). Earlier CoA in the dominant leg stance phase appears to be a consequence of the reduction in activation during the second peak of the activation profile (Fig 8). This reduction in activation would reflect a decrease in muscle activity related to push-off and possibly reflects a more cautious gait. The earlier CoA of the activation profile associated with heel-strike reflects a more sustained activation following a slower build-up (Fig 8). Again, this may be related to active control over CoM movement during the stance phase or a more cautious weight acceptance. In the supplementary material we reported the Falls Efficacy Scale International (FES-I) results measured at Pre, Post2, and retention (2 weeks after the last training session) time-points. FES-I decreased between Pre and retention time-points, and between Pre and Post2, however insignificant, but the trend suggested increasing confidence, which may have contributed to a less cautious behaviour (S1 File). The active control of CoM movement is supported by the fact that muscles that would contribute to mediolateral control, specifically tibialis anterior, peroneus longus, and gluteus medius are part of these synergies. To check that changes in CoA and FWHM of the activation profiles were not due to changes in the duration of gait phases, we assessed single support and double support times as percentages of the stride times, and no effects of Condition were found.

## Limitations

There are several limitations in our study that need to be addressed. First, we did not include a young group to investigate the influence of aging on the transfer effect of standing balance training to gait. Second, we did not include a control group to identify the normal variation over time, independent of the training program. Third, it was hypothesized that transfer would occur through shared synergies. In the current study muscle weightings of the synergy analysis were extracted based solely on gait data rather than a combination with standing balance data. Therefore, it is less likely that any one of these synergies is optimally defined as being a shared synergy between the tasks. Moreover, we included only 11 muscles in the synergy analysis not representative of whole-body activity. It could be that the mechanical changes

observed would be more closely associated with changes in control of muscles in the upper body [75]. We extracted five synergies to describe leg muscle activity across both narrow-base and normal walking, together accounting for 87% of the variation in muscle activity. In spite of differences in muscles measured, participant age, and walking conditions between studies, (the number and the general grouping and activation of) these synergies resembled results reported previously [26,76–80]. We kept the muscle weightings in these synergies constant over conditions and time-points to investigate variations in the activation profile. We repeated the analysis for six synergies to achieve a variance accounted for over 90%. However, the results were not any different.

## Conclusions

In conclusion, after ten sessions of standing balance training, older adults decreased their step width in narrow-base walking and decreased the step width variability in narrow-base and normal. This suggests a transfer of balance skills from standing to walking after ten training sessions. However, there was no evidence in adaptations of neuromuscular control due to balance training associated with changes in control of mediolateral gait stability. In addition, older adults adapted mediolateral CoM kinematics and the step width between normal and narrow-base walking, and this was associated with changes in synergies governing the activation of leg muscles. Our results suggest that in older population the neural mechanisms are still adaptable and acquired skills can be transferred from standing to walking.

## Supporting information

**S1 File.**
(DOCX)

**S2 File.**
(MAT)

**S3 File.**
(CSV)

**S4 File.**
(CSV)

**S5 File.**
(MAT)

## Acknowledgments

The research team would like to thank the individuals who participated in the experiment.

## Author Contributions

**Conceptualization:** Leila Alizadehsaravi, Sjoerd M. Bruijn, Jaap H. van Dieën.

**Data curation:** Leila Alizadehsaravi, Wouter Muijres, Ruud A. J. Koster.

**Formal analysis:** Leila Alizadehsaravi.

**Funding acquisition:** Jaap H. van Dieën.

**Investigation:** Leila Alizadehsaravi, Wouter Muijres, Ruud A. J. Koster.

**Methodology:** Leila Alizadehsaravi, Sjoerd M. Bruijn, Wouter Muijres, Ruud A. J. Koster.

**Project administration:** Leila Alizadehsaravi, Jaap H. van Dieën.

**Resources:** Jaap H. van Dieën.

**Software:** Leila Alizadehsaravi.

**Supervision:** Sjoerd M. Bruijn, Jaap H. van Dieën.

**Validation:** Leila Alizadehsaravi, Sjoerd M. Bruijn, Jaap H. van Dieën.

**Visualization:** Leila Alizadehsaravi.

**Writing – original draft:** Leila Alizadehsaravi, Ruud A. J. Koster, Jaap H. van Dieën.

**Writing – review & editing:** Leila Alizadehsaravi, Sjoerd M. Bruijn, Wouter Muijres, Ruud A. J. Koster, Jaap H. van Dieën.

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
