## [Decision Letter · Decision Letter 0]

10 Feb 2021

PONE-D-20-30116

Neuromuscular control of gait stability in older adults is adapted to environmental demands but not improved after standing balance training

PLOS ONE

Dear Dr. van Dieën,

Thank you for submitting your manuscript to PLOS ONE. After careful consideration, we feel that it has merit but does not fully meet PLOS ONE’s publication criteria as it currently stands. Therefore, we invite you to submit a revised version of the manuscript that addresses the points raised during the review process.

Firstly, sorry for the slow review process, but I had to submit for a third reviewer. One reviewer suggested to reject the paper. The other two reviewers were favorable with the publication of the paper, but many points need to be improved. Please address all suggestions of the three reviewers. Please try to address the suggestions of the reviewer that indicated to reject the paper too (Reviewer 2). There are some relevant aspects in the review. For example, the rationale that training standing balance would improve walking balance needs to be improved. Also, I ask special attention in the discussion to explain the absence of effect of the balance training - both reviewers have indicated that this aspect should be better explained.

We look forward to receiving your revised manuscript.

Kind regards,

Fabio A. Barbieri, PhD

Academic Editor

PLOS ONE

Reviewers' comments:

Reviewer's Responses to Questions

**Comments to the Author**

1. Is the manuscript technically sound, and do the data support the conclusions?

Reviewer #1: Yes

Reviewer #2: No

Reviewer #3: Yes

2. Has the statistical analysis been performed appropriately and rigorously? 

Reviewer #1: Yes

Reviewer #2: No

Reviewer #3: No

3. Have the authors made all data underlying the findings in their manuscript fully available?

Reviewer #1: Yes

Reviewer #2: Yes

Reviewer #3: Yes

4. Is the manuscript presented in an intelligible fashion and written in standard English?

Reviewer #1: Yes

Reviewer #2: No

Reviewer #3: Yes

5. Review Comments to the Author

Reviewer #1: The study investigated the skill transfer of standing balance training to walking balance among older adults. Walking balance was tested in normal and narrow-base walking conditions. The findings quantified a series of gait measures including muscle synergies, COM displacement, and gait stability. The study had a null finding on the training effect on gait despite positive effects of training on stance balance reported in an earlier study using the same cohort of participants. The study is generally well-done, and the results are relevant to both researchers and clinicians. Addressing the issues noted below will increase the clarity of the manuscript.

Major issues:

1. Interpretation of null effect: The authors speculate that the duration of training was the cause of the null effect. I disagree with this cause for two reasons. First, if the duration was too short, I would have expected to see some improvement between post 1 and post 2. Second, there is a significant effect of training on standing balance from your previous paper (ref#24, Alizadehsaravi et al. 2020). If the duration was sufficient for standing, what is the rationale that it was not sufficient for walking?

2. Please justify the selection of muscles. Specifically, muscles that were examined were mostly sagittal, but the task is more medial-lateral than sagittal, so perhaps it is not surprising that sagittal muscles synergies were not affected by the training. Gluteus medius is the only muscle that acts mainly in the ML plane (hip abductor).

3. Recommendation for the structure of the discussion. The following text: “The gait measures were different between conditions but not affected by training” was emphasized several times in the discussion, and demonstrates that these two issues were conflated throughout. It would increase clarity to separate the discussion for the gait adaptation from the discussion for training. The adaptation is shown in several different gait measures and, in my opinion, is the most interesting finding of this study. I suggest the authors have stand-alone discussion paragraph(s) focusing on changes in gait control for narrow base walking and what they would mean in real life locomotion and falls. More in-depth discussion should address the questions such as, are all the changes in gait control adaptation to the postural challenge? Could they be maladaptive or mechanical consequences of a narrower base of support?

4. This is not necessary, but it could be of interest: Would it be possible to quantify the synergies for standing balance? In lines 58-59 the authors note that four synergies are shared between walking and reactive balance control. Adding the standing balance synergies to the current paper on gait synergies would provide a more comprehensive study.

5. Please specify the type of synergy analysis that was conducted, for example, principal component analysis, UCM analysis, or other approach.

Minor comments:

1. Line 34 “Falls in older adults mostly occur during walking. Therefore, skills acquired during stance balance should transfer to gait and improve gait stability”. The logic and/or the wording is not clear across the first two sentences, it is explained more fully in later sentences, but the second sentence should not start with “Therefore,…”.

2. Line 51 and line 276 - Confidence is mentioned in a couple of locations in the manuscript, but confidence was not measured in the current study. Perhaps it should be noted as a need for future study.

3. Line 105-108 – Please explicitly describe the exercise plan, including progressions and progression criteria, so that the reader can determine if the training was sufficiently challenging and was appropriately progressed. Note that the progressions were not available in previous publication either. The null effects can possibly be explained by these training details.

4. Since the journal does not have a word limit, I recommend that more details of the methods are included in the current paper, rather than simply directing readers to the previous paper for the method section (Alizadehsaravi et al., 2020). For example, familiarization trials mentioned in figure 1 can also be explained in the text. Rationales could be provided for the treadmill settings (4.5 minutes at a constant speed of 3.5 km/h).

5. Foot placement error was reported to indicate the narrow base walking performance. The mean error is less than 2 cm, which is presumably quite small. Could the training be subject to a ceiling effect? It could be helpful to also report step width and step width variability, to aid the interpretation of narrow base walking performance in comparison with normal walking.

Editorial comments:

Line 90 - Exclusion criteria 4 minutes without an aid – in previous paper it is three minutes.

Line 211 typo “did not”

Figures - Be consistent with the scales, number of decimal points – use different markers for normal and narrow gait, to accommodate people without color printers.

Figure 6 – Authors should provide the x-axis label. Error bars description?

Line 227 – “87­+2%”. Consistent with spacing.

Reviewer #2: The current manuscript reports the results of a study that sought to examine whether standing balance training altered or improved balance performance and/or its control in older adults. Measures derived from the center of mass (i.e., displacement, velocity, LDE) and foot placement were used to characterize performance, while a host of muscle recruitment and coordination metrics are employed to characterize the control of balance. While further efforts to understand the neuromuscular basis for walking and balance control, as well as the effect of training or aging, are always welcome, my enthusiasm for the current manuscript is dampened by issues regarding study motivation and scientific rationale, technical/methodological choices, and the lack of a discussion that explores, explains, or puts the current results in content with previous research.

Major

Study motivation and scientific rationale

1. It is unclear to me why training in standing balance would have an effect on the control or performance of walking balance. This would seem to go against the principle of specificity, which has been frequently described and cited in the balance literature (Grabiner; Oddsson). Additionally, multiple studies have shown standing balance does not equate to walking balance (Mackey, 2005; Owings, 2000). It is therefore unclear what evidence would support the idea that training standing balance would improve walking balance or alter its control.

References

Oddsson et al., 2007. How to improve gait and balance function in elderly individuals—compliance with principles of train.

Grabiner et al., 2012. Task-specific training reduces trip-related fall risk in women.

Mackey et al., 2005. Postural steadiness during quiet stance does not associate with ability to recover balance in older women.

Owings et al., 2000. Measures of postural stability are not predictors of recovery from large postural disturbances in healthy older

2. The lack of scientific rationale is reflected in the lack of structure and organization in the introduction. As currently written, each paragraph in the introduction does not build on the previous paragraph in a way that creates a logical argument and directs the reader to an obvious gap/need (i.e., study objective), and an accompanying hypothesis. In fact, no specific statement of purpose and/or testable hypotheses are provided. Similarly, the contents within each introduction paragraph do not support or back up the topic sentence at the start of each paragraph. Greater attention to "building your argument" based on the published literature is required in the introduction.

Technical issues

3. It is not clear why the extraction of muscle synergies was limited to five. A number of studies have shown the number of muscle synergies to change with skill or training. This methodological choice would therefore seem at odds with the goal of the study; to examine whether training alters neuromuscular control of walking balance. It would seem more suitable to identify the number of muscle synergies for each subject in each condition that is required to explain the original EMG.

4. The extracted synergies are reported to account for 85% of the variance in the EMG signals. Is this sufficient? Typically, studies that extract muscle synergies from EMG signals use between 90 or 95% variance accounted for. Similarly, was the variance accounted in a specific muscle, across all conditions? Additional explanation and justification for how and why muscle synergies were extracted in the manner they were is required.

5. Parametric inferential statistics are used to compare conditions and examine the effect of training. However, no assessments of normality were performed or reported. Please provide the results of Shapiro-Wilk's tests if parametric statistics are to be used.

6. Little information is provided about study participants. Did the study participants have a history of falling? High or low balance confidence or falls self-efficacy scores? In the absence of additional information describing the balance and falls characteristics of the study participants, it is difficult to interpret the study results and put them in context.

Discussion

7. As currently written, the discussion largely reiterates the study results. At present there is no explanation for why training failed to have an effect on walking balance performance or control. This seems critical as it appears to be the primary motivation for the study. Also, there is no discussion of how the current study results regarding muscle synergies and balance control compare to or differ from studies previously conducted (e.g., da Silva Costa, 2020; Allen, 2020; Monaco, 2010). As a result, it is not clear what additional insight is provided by the current study to advance our understanding of balance control and aging. A major revision to the discussion is likely required.

Minor

1. Please review the manuscript for grammatical and spelling errors.

2. It is unclear why gait speed was kept constant between walking conditions (regular and narrow), between participants, and between time points (i.e., pre and post). Would walking speed not be one of the variables expected to improve with training? Please provide justification for both the selection of this specific gait speed, as well as its constraint across and within study conditions and time points.

3. Results: the section describing the spatial components of muscle synergies reads more like a discussion or interpretation of the spatial patterns. Consider revising this section to simply report rather than interpret the results.

4. Figure 1 is not required. The section in the methods provides sufficient information on the study protocol.

5. Figures 3-5, and 8 are very difficult to interpret. I would advise removing the lines connecting each of the phases of the study.

6. Justification for the selection of the wide range of balance performance and control metrics are required. Upon what basis did the authors expect these metrics to change as a function of training?

7. Additional details regarding the standing balance training and subsequent results are required. Perhaps the training procedures could be included in an Appendix. The results of the standing training should be briefly summarized in the results to provide the reader with context.

Reviewer #3: This manuscript describes a study investigating the effect of balance training on neuromuscular and stability control in older adults during normal and narrow base walking. The findings indicated no training effects on these parameters. As expected, the base of support constraint resulted in walking adaptations in both neuromuscular and stability parameters. Although the manuscript is well organized, some aspects could be improved to facilitate readability to those not familiarized with some parameters analyzed in this study. The discussion could also be improved to explain the absence of effect of the balance training.

1) The abstract would benefit from the use of plain language, particularly when describing the results. The use of jargon (particularly FWHM and CoA) makes it difficult to follow and understand the main findings.

2) The first two sentences of the introduction suggest a logical interconnection that is not the case. The fact that people often fall during walking does not imply that a “balance training should transfer to gait and improve gait stability”.

3) How was the sample size determined for this study? Was there any sample size calculation?

4) The description of the balance training protocol (page 6, lines 104-109) is very brief. It is essential to provide additional information regarding the duration of each session, the progression of the exercises throughout the sessions, and the organization of each session (warm-up, main activities, etc.).

5) When describing the FWHM and CoA (page 9, lines 186-192), include the meaning of these variables for those not familiarized with these variables.

6) For the beam walking task, did the participants step off the beam at any moment? What was the maximum foot placement error accepted when the participant eventually did not step with the entire foot over the beam?

7) Since the beam width (12 cm) is remarkably close to the average foot width, how foot placement error could be affected by balance training? Wouldn’t you have a floor effect for this variable?

8) Page 13 (lines 286-287): what do you mean by “robustness of gait”?

9) The discussion about the absence of effect of the balance training in all investigated variables was incredibly brief (page 14, lines 313-317). Most of the discussion is focused on the effects of the base of support manipulation. The discussion regarding the balance training needs to improve to explain the absence of an effect. Only arguing that the duration of the training protocol was short is not enough. Why did you choose a 3-week intervention if studies are showing that more time is needed to observe gait improvements? Were the variables not sensitive enough to capture gains due to balance training?

10) I recommend including a paragraph with the limitations of this study.

6. PLOS authors have the option to publish the peer review history of their article (what does this mean?). If published, this will include your full peer review and any attached files.

Reviewer #1: No

Reviewer #2: No

Reviewer #3: No

---

## [Author Response · Author response to Decision Letter 0]

8 Mar 2022

Reply to reviewers

We would like to thank the reviewers for their comments and suggestions, which have helped improve this paper. The introduction and discussion sections of the manuscript have been modified extensively. Below we have copied the reviewers' comments and have provided a point-by-point reply.

Reviewer #1: The study investigated the skill transfer of standing balance training to walking balance among older adults. Walking balance was tested in normal and narrow-base walking conditions. The findings quantified a series of gait measures, including muscle synergies, COM displacement, and gait stability. The study had a null finding on the training effect on gait despite positive effects of training on stance balance reported in an earlier study using the same cohort of participants. The study is generally well-done, and the results are relevant to both researchers and clinicians. Addressing the issues noted below will increase the clarity of the manuscript.

Major issues:

1- Interpretation of null effect: The authors speculate that the duration of training was the cause of the null effect. I disagree with this cause for two reasons. First, if the duration was too short, I would have expected to see some improvement between post 1 and post 2. Second, there is a significant effect of training on standing balance from your previous paper (ref#24, Alizadehsaravi et al. 2020). If the duration was sufficient for standing, what is the rationale that it was not sufficient for walking?

We thank the reviewer for this comment. Following the reviewer’s suggestions in minor comment #5, step width and step width variability have been added to the gait kinematics analysis. Based on these new analyses we did find a positive effect of the balance training on gait. The transfer effect was evident in decreased step width and decreased step width variability after three weeks of standing balance training. Note that the primary measures, defined as the foot placement error and the percentage of the steps within the beam, did not change after the training. Therefore, we conclude that transfer did happen, but some caution is needed as this was not demonstrated by our primary outcome variable. The introduction and methods have been updated to include step width and step width variability and the discussion has been modified extensively.

2- Please justify the selection of muscles. Specifically, muscles that were examined were mostly sagittal, but the task is more medial-lateral than sagittal, so perhaps it is not surprising that sagittal muscles synergies were not affected by the training. Gluteus medius is the only muscle that acts mainly in the ML plane (hip abductor).

We thank the reviewer for this comment. We agree with the reviewer that most of the measured muscles were not acting solely in the frontal plane. Although we focused on balance in the frontal plane, motions in all three planes are important for a stable gait pattern. Moreover, muscles do not have effects in one plane, but produce effects in multiple planes. For example, sagittal plane push-off, which is mainly generated by the calf muscles, acts at some distance to the mid-sagittal plane through which push-off modulation can contribute to frontal plane balance control. We measured almost all possible ankle muscles as they play a key role in postural stability 4–6. A simulation study also showed that in the late stance of a gait cycle, the gluteus medius rotates the body towards the ipsilateral leg, while the soleus and gastrocnemius rotate the body towards the contralateral leg4. 

Biceps femoris, rectus femoris, and vastus lateralis were measured as they contribute to stabilizing the knee around heel strike and in single leg stance.

Moreover, we chose similar muscles to a previous study that showed changes in synergies due to long-term training during walking3. We also measured the adductor longus, but we omitted it from the analysis in view of the signals being very noisy, especially in female subjects. 

We now provide this motivation in the manuscript Lines 194-197.

3- Recommendation for the structure of the discussion. The following text: "The gait measures were different between conditions but not affected by training" was emphasized several times in the discussion, and demonstrates that these two issues were conflated throughout. It would increase clarity to separate the discussion for the gait adaptation from the discussion for training. The adaptation is shown in several different gait measures and, in my opinion, is the most interesting finding of this study. I suggest the authors have stand-alone discussion paragraph(s) focusing on changes in gait control for narrow base walking and what they would mean in real life locomotion and falls. More in-depth discussion should address the questions such as, are all the changes in gait control adaptations to the postural challenge? Could they be maladaptive or mechanical consequences of a narrower base of support?

We thank the reviewer for the suggestions. The discussion has been updated and restructured to general findings, the effect of narrow base, transfer effect, limitations, and conclusions, to address this suggestion. Additionally, lines 465-471 have been added to the discussion (effects of narrow base) to focus on changes in gait control for narrow base walking and what it means in real life. 

About the last part of the reviewer's comment, we do not expect the modulation of the gait parameters between the narrow base and normal gait to be maladaptive because the decreased center of mass displacement and velocity are beneficial for balance control. Note that in a challenging condition the efficiency-safety trade-off moves more to the side of safety. However, the adaptation to the mechanical constraints that we imposed will have mechanical consequences. It was our aim to understand the neural responses needed to successfully deal with these mechanical constraints.

Since in our new analysis we found the effect of training to be transferred to gait, we have rewritten the discussion according to the new findings. There are separate paragraphs discussing the adaptation and the training effect as the reviewer suggested. 

4- This is not necessary, but it could be of interest: Would it be possible to quantify the synergies for standing balance? In lines 58-59 the authors note that four synergies are shared between walking and reactive balance control. Adding the standing balance synergies to the current paper on gait synergies would provide a more comprehensive study.

We thank the reviewer for the comment. The paper we referred to in the introduction showed that most muscle synergies used in perturbation responses during standing were also used in perturbation responses during walking, suggesting common neural mechanisms for reactive balance across different contexts. (DOI: 10.3389/fncom.2013.00048)

The reviewer's suggestion is indeed interesting. However, to make a fair comparison of the synergies we would need to use the same set of muscles in standing and in gait. Our balance study was unipedal (unlike the study we referred to), which will make a significant difference in the grouping of muscles into synergies. Therefore, we decided not to include the synergies of standing balance in this study. We have analyzed the synergies of stance leg and trunk muscles and have observed training effects on perturbed standing balance, which is reported elsewhere (DOI: https://doi.org/10.1101/2021.03.31.437824).

5- Please specify the type of synergy analysis that was conducted, for example, principal component analysis, UCM analysis, or other approach.

We thank the reviewer for the comment. We have now more clearly written that we use NNMF to extract synergies;

“Synergies were extracted from 11 muscles using non-negative matrix factorization based on Lee and Seung's multiplicative update rule7 with 50 repetitions with a maximum of 1000 iterations to update the components and at a tolerance of 10-6.”Lines 245-247.

Minor comments:

1- Line 34 "Falls in older adults mostly occur during walking. Therefore, skills acquired during stance balance should transfer to gait and improve gait stability". The logic and/or the wording is not clear across the first two sentences, it is explained more fully in later sentences, but the second sentence should not start with "Therefore,…".

We thank the reviewer for this comment. We noted the suggestion, and the manuscript has been updated as follows:

“Falls in older adults mostly occur during walking [1,2]. Thus, if skills acquired by balance training programs do not transfer to improvements in gait stability, they are unlikely to decrease the number of falls.” Lines 37-39.

2- Line 51 and line 276 - Confidence is mentioned in a couple of locations in the manuscript, but confidence was not measured in the current study. Perhaps it should be noted as a need for future study.

We thank the reviewer for the comment. We know that older adults take wider step in order to increase their base of support and to be robust against the perturbations, which can be regarded as cautious behavior. Based on our new results, the decreased step width and decreased step width variability after the longer duration of training, we concluded that the improved gait biomechanics may reflect an increased confidence. This has been further discussed in the discussion section.

We have measured the concern of falls using the FESI questionnaire, but so far did not report it. We have updated the manuscript's supplementary materials to include the FESI scores. Between Pre and Post2, although there was a trend, there was no significant decrease in FESI score. However, at retention, the FESI score had decreased significantly, implying that weeks after training, participants felt more confident about their balance ability. The results are shown below:

"A repeated measures ANOVA indicated that concern of falling was affected by balance training. Post-hoc analysis showed that concern of falling was not significantly changed immediately after the training program but was decreased at the retention measurement 2 weeks after the end of training (t = 2.16, p = 0.072; t = 2.82, p = 0.022, respectively."

Figure 1. FES-I scores at different time points. Each of the lines between timepoints represents the score of a single participant.

3- Line 105-108 – Please explicitly describe the exercise plan, including progressions and progression criteria, so that the reader can determine if the training was sufficiently challenging and was appropriately progressed. Note that the progressions were not available in previous publications either. The null effects can possibly be explained by these training details.

We thank the reviewer for the comment. We have provided more information on the training program and the training materials as supplementary material. Progression criteria were based on the physical therapist's observation during the training sessions; if participants were able to perform the task for 60 seconds, the difficulty level was increased. The progression was hard to measure objectively as the training was group-based. But the general progression plan was as follows:

Table 1. Guideline for training progression

Number Exercise Duration/Frequency

Warm-up

1) Head rotations Rotate head to either side 5 x

3 repetitions 

2) Back stretching Stretch 3 x

3 repetitions

3) Trunk rotations 5 rotations to both sides

3 repetitions

Exercises

4) Balancing 

- One leg stance (when possible)

- Switch the legs 

- Unstable surfaces 3 x 60 seconds

2 repetitions

5) Balancing eyes-closed 

- One leg stance (when possible)

- Switch the legs 

- Unstable surfaces 3 x 60 seconds

2 repetitions 

6) Displacement of weight

- One leg stance

 - Switch the legs 

- Unstable surfaces 3 x 60 seconds

2 repetitions 

7) Passing/throwing around a ball in groups of 4

Fitness ball

- One leg

- Unstable surface

2 kg ball

- One leg

- Unstable surface

Alternative approaches:

- Make the circle bigger.

 - With back towards each other in order to induce more trunk rotations. 5 rounds both directions

3 repetitions

8)

 Pass the big ball around while stopping it on foot and role it to the other person.

Fitness ball

- One leg

- Unstable surface

2 kg ball

- One leg

- Unstable surface 5 rounds both directions

3 repetitions

Figure. 2. Balance training materials 

Note that exercises that specifically target gait were excluded from the training to investigate the transfer effect.

4- Since the journal does not have a word limit, I recommend that more details of the methods are included in the current paper, rather than simply directing readers to the previous paper for the method section (Alizadehsaravi et al., 2020). For example, familiarization trials mentioned in figure 1 can also be explained in the text. Rationales could be provided for the treadmill settings (4.5 minutes at a constant speed of 3.5 km/h).

We thank the reviewer for the comment. We have included the following additional information in the methods section:

“To quantify transfer to gait, participants were instructed to walk for 4.5 minutes on a treadmill with an embedded force plate. For estimating the local divergence exponent, a minimum of 150 steps is recommended [41]. We expected that a total duration of 4.5 minutes was needed to reach that number. To avoid effects of gait speed on outcome measures, this was kept constant at 3.5 km/h for all participants [42].” Lines 165-169.

“A narrow-base walking paradigm was chosen because narrow-base walking has been shown to challenge mediolateral stability in older adults [12,34].” Lines 174-175.

“Participants were acquainted with the experimental setup to minimize habituation effects. For familiarization, participants performed 30 seconds of narrow-base walking before the measurement.” Lines 178-180.

5- Foot placement error was reported to indicate the narrow base walking performance. The mean error is less than 2 cm, which is presumably quite small. Could the training be subject to a ceiling effect? It could be helpful to also report step width and step width variability, to aid the interpretation of narrow base walking performance in comparison with normal walking.

We thank the reviewer for this valuable comment. The suggestion has made us do further analysis and we found that step width and step width variability were decreased after training. These new findings do indicate a transfer effect of balance training to gait. The new results have been added to the manuscript. Regarding a possible ceiling in the mean step error, one should consider that the width of the beam is 12 cm, whilst the average foot width is about 10 cm (Jurca, Zabkar, Dzeroski, 2019). This means that participants, on average, had a 2 cm margin in foot placement that was not seen as error. So, the task did not require perfect control. Additionally, the percentages of steps inside the beam were 27.6% (SE 3.9%) pre-training and 33.3% (SE 4.4%) post-training. We do believe this suggests that there is ample room for improvement. The best recorded performance was 67% of steps within the beam, which demonstrated to us that scores much higher than the mean could be obtained, and that even these best performances were not close to the end of the scale.

Editorial comments:

1- Line 90 - Exclusion criteria 4 minutes without an aid – in the previous paper it is three minutes.

Thank you for the comment. Indeed, it was three minutes. The manuscript has been corrected. Line 140.

2- Line 211 typo "did not"

The repetition of "did not" has been removed. Thank you for noticing.

3- Figures - Be consistent with the scales, number of decimal points – use different markers for normal and narrow gait, to accommodate people without color printers.

Thank you for the comment. We double-checked the manuscript and have corrected the figures as suggested. 

4- Figure 6 – Authors should provide the x-axis label. Error bars description?

Thank you for the comment. The error bars were not needed and have been removed, thank you. 

The X-axis label was an arbitrary unit and has been added to the revised version. 

5- Line 227 – "87­+2%". Consistent with spacing.

Thank you for the comment. We have corrected the manuscript and put spaces in between as follows: 87 ± 2% 

Reviewer #2: The current manuscript reports the results of a study that sought to examine whether standing balance training altered or improved balance performance and/or its control in older adults. Measures derived from the center of mass (i.e., displacement, velocity, LDE) and foot placement were used to characterize performance, while a host of muscle recruitment and coordination metrics are employed to characterize the control of balance. While further efforts to understand the neuromuscular basis for walking and balance control, as well as the effect of training or aging, are always welcome, my enthusiasm for the current manuscript is dampened by issues regarding study motivation and scientific rationale, technical/methodological choices, and the lack of a discussion that explores, explains, or puts the current results in content with previous research.

Major

Study motivation and scientific rationale

1- It is unclear to me why training in standing balance would have an effect on the control or performance of walking balance. This would seem to go against the principle of specificity, which has been frequently described and cited in the balance literature (Grabiner; Oddsson). Additionally, multiple studies have shown standing balance does not equate to walking balance (Mackey, 2005; Owings, 2000). It is therefore unclear what evidence would support the idea that training standing balance would improve walking balance or alter its control.

References

Oddsson et al., 2007. How to improve gait and balance function in elderly individuals—compliance with principles of train.

Grabiner et al., 2012. Task-specific training reduces trip-related fall risk in women.

Mackey et al., 2005. Postural steadiness during quiet stance does not associate with ability to recover balance in older women.

Owings et al., 2000. Measures of postural stability are not predictors of recovery from large postural disturbances in healthy older

We thank the reviewer for the references. Based on the first reviewer’s suggestion we have added two new measures to our gait evaluation. We found decreased step width in narrow-base and decreased step width variability in both conditions after training. Therefore, we argue that there was a transfer effect from standing balance to narrow-base walking, although it was not visible in the primary outcome measures, perhaps due to the lower sensitivity of those measures.

The articles suggested by the reviewer were reviewed carefully. 

Indeed, the reviewer is right, that in general, training is very task specific. However, as outlined in our introduction, studies that trained standing balance found reduced falls, and falls in daily life often happen during walking. We have now further clarified why transfer effects may be expected in lines 41-44 and lines 71-80.

2- The lack of scientific rationale is reflected in the lack of structure and organization in the introduction. As currently written, each paragraph in the introduction does not build on the previous paragraph in a way that creates a logical argument and directs the reader to an obvious gap/need (i.e., study objective), and an accompanying hypothesis. In fact, no specific statement of purpose and/or testable hypotheses are provided. Similarly, the contents within each introduction paragraph do not support or back up the topic sentence at the start of each paragraph. Greater attention to "building your argument" based on the published literature is required in the introduction.

Thank you for the comment. We have completely reworked the introduction, and made sure to link the paragraphs, and provide a better rationale for our study. 

Technical issues

3- It is not clear why the extraction of muscle synergies was limited to five. A number of studies have shown the number of muscle synergies to change with skill or training. This methodological choice would therefore seem at odds with the goal of the study; to examine whether training alters neuromuscular control of walking balance. It would seem more suitable to identify the number of muscle synergies for each subject in each condition that is required to explain the original EMG.

Thank you for the comment. Several authors have reported the number of synergies during gait, and most have reported 4 to 6 synergies. We determined the number of synergies from the concatenated data of all three-measurement timepoints. So, this constrains the number of synergies as well as the muscle weighting factors to be constant across timepoints, while differences in activation profiles between pre- and post-training can be observed. Additionally, having a different number of synergies (i.e., not constraining muscle weighting factors over time points) and different activation profiles will not allow a conclusive result regarding changes due to training. The problem then is that any difference in muscle weightings might as well be explained by different activation profiles (and vice versa), rather than training. 

4- The extracted synergies are reported to account for 85% of the variance in the EMG signals. Is this sufficient? Typically, studies that extract muscle synergies from EMG signals use between 90 or 95% variance accounted for. Similarly, was the variance accounted in a specific muscle, across all conditions? Additional explanation and justification for how and why muscle synergies were extracted in the manner they were is required.

Thank you for the comment. Several other studies have also used an 85% threshold [9][10] Considering the number of muscles measured, 5 synergies is an acceptable number, and adding more synergies wouldn't add much value to the current reconstructed EMG. To check for this, we have added one more synergy (see figure below) while VAF improved to >90%, it was only 4% higher with 6 than with 5 synergies. 

Unfortunately, there are only a few studies on synergies in older adults that we could refer to, but considering the lower signal to noise ratio of EMG data in older adults, lowering the threshold to 85% for reconstructing the data is not unreasonable (as shown by Baggen et al.[9]), and using 90% might cause overfitting. There are no standards for choosing the threshold or the method, but our choices were based on our research questions and our target group.

Figure 2. decomposed synergies to temporal and spatial components, with 6 (top) and 5 (bottom) synergies

5- Parametric inferential statistics are used to compare conditions and examine the effect of training. However, no assessments of normality were performed or reported. Please provide the results of Shapiro-Wilk's tests if parametric statistics are to be used.

Thank you for the suggestion. Shapiro-Wilk’s test has now been performed and in two of our measures (CoM variability and vCoM) data were not normally distributed. Therefore, log transformation was performed before applying the repeated measure ANOVA. The results were not changed. The Shapiro-Wilk has also been applied to the two new measures (step width and step width variability).

6- Little information is provided about study participants. Did the study participants have a history of falling? High or low balance confidence or falls self-efficacy scores? In the absence of additional information describing the balance and falls characteristics of the study participants, it is difficult to interpret the study results and put them in context.

The participants in our study had no history of falls in the year prior to the measurement. The methods section for participants has been expanded as below: 

“Twenty-two older (72.6 ± 4.2 years old; mean ± SD, 11 females) healthy volunteers, without a history of falls in the preceding year, participated in this study. The required sample size was estimated at twenty-two based on power analysis for an F test of a repeated measure, assuming an effect size of 0.44 and correlation among repeated measures of 0.6 (β = 0.8, G * power 3.1.9.2, Düsseldorf, Germany), comparable to similar studies [38,39].” Lines 132-135. 

Balance performance of these participants was a primary outcome. This was extensively reported in a previous paper, but results are briefly reported as well as in this paper. FESI scores are provided in the supplementary material.

Discussion

7- As currently written, the discussion largely reiterates the study results. At present there is no explanation for why training failed to have an effect on walking balance performance or control. This seems critical as it appears to be the primary motivation for the study. Also, there is no discussion of how the current study results regarding muscle synergies and balance control compare to or differ from studies previously conducted (e.g., da Silva Costa, 2020; Allen, 2020; Monaco, 2010). As a result, it is not clear what additional insight is provided by the current study to advance our understanding of balance control and aging. A major revision to the discussion is likely required.

We thank the reviewer for this critical comment. The new findings of training effects for step width and step width variability forced us to rewrite the whole discussion.

We tried to make it clear in the introduction that the novel aspects of our study are that: 1) we addressed balance training and transfer to gait in older adults, 2) we quantified the effect of short and longer duration of training, 3) we solely focused on standing balance training. We used unstable balance training and excluded exercises that targeted gait stability directly, to focus on transfer of balance skill to gait as a result of standing balance training. This suggests the effectiveness of the standing balance training in older adults, providing a clue for clinicians and direction for future studies, that standing balance training may be valuable for older adults with declined mobility. We have rewritten the discussion to reflect these ideas more clearly. The idea behind it is if the standing balance training can be transferred to gait, there might be a potential clue for clinicians to use standing balance training to improve gait postural control in less mobile older adults. 

Minor

1- Please review the manuscript for grammatical and spelling errors.

Thank you for the comments. The manuscript has been checked and corrected. 

2- It is unclear why gait speed was kept constant between walking conditions (regular and narrow), between participants, and between time points (i.e., pre and post). Would walking speed not be one of the variables expected to improve with training? Please provide justification for both the selection of this specific gait speed, as well as its constraint across and within study conditions and time points.

Thank you for the comment. Our rationale to use a constant gait speed is that we were primarily interested in the effect of balance training on gait stability, and not specifically on gait speed, while speed is known to affect stability. Our main focus was on effects on the quality of balance control, which would be reflected in changes in stability measures (LDE, vCoM) and task performance measures (foot placement error, % steps in beam). Had we not constrained gait speed, but taken it as an additional outcome measure, it would have become a confounder in the analysis of all other variables. We have more explicitly stated this in the manuscript: “To avoid effects of gait speed on outcome measures, this was kept constant at 3.5 km/h for all participants.” Lines 168-169.

3- Results: the section describing the spatial components of muscle synergies reads more like a discussion or interpretation of the spatial patterns. Consider revising this section to simply report rather than interpret the results.

Thank you for the comment. We have updated the manuscript as suggested.

4- Figure 1 is not required. The section in the methods provides sufficient information on the study protocol.

Thank you for the suggestion. We prefer to keep the figure to help grasping the experimental procedure in one glance.

5- Figures 3-5, and 8 are very difficult to interpret. I would advise removing the lines connecting each of the phases of the study.

Thank you for the comment. We have updated the figures. 

6- Justification for the selection of the wide range of balance performance and control metrics are required. Upon what basis did the authors expect these metrics to change as a function of training?

Thank you for the comment. 

Performance measure: Mean foot placement error

This measure was primarily used, rather than comparable measures of medio-lateral foot placement (e.g. step width), because it is by design the most direct quantification of task performance. Hence, we expected it to be the most sensitive measure to changes in task performance.

Kinematics derived: Trunk CoM displacement, trunk CoM velocity, and Local divergence exponent (LDE)

The first two are the standard and most widespread used measures to express gait stability. However, particularly for use during gait these measures are not entirely undisputed as they are not necessarily minimized for task execution. On one hand, relatively minor CoM movement might indicate good motor control and, therefore, good balance ability. On the other hand, relatively large CoM movement might indicate that it is not minimized because of good balance ability.

Because aforementioned metrics leave room for interpretation the choice was made to add another type of measure with proven validity for gait. The choice for the Lyapunov exponent as an outcome measure was based on the existence of substantial evidence suggesting its validity in the context of gait stability and falling [4].

Improved performance and stability of gait would directly express transfer of training at the level of task performance.

As described above we expected changes in gait performance to result from improved control, which we assessed in terms of muscle synergies that could be expected to be common to control of stance and gait. To quantify changes in these synergies we used the full width at half maximum (FWHM) and Centre of activation (CoA). The FWHM metric addresses the duration of activation but is naïve of timing of activation. The CoA metric addresses the timing of activation but is naïve of duration of activation. By including both we get the entire picture. These measures have been shown to be sensitive to task challenge and thus we expected them to be sensitive also to changes in capacity of our subjects to control their balance in a given task.

These justifications for the selection of measures have been added to their corresponding part in the method section. 

7- Additional details regarding the standing balance training and subsequent results are required. Perhaps the training procedures could be included in an Appendix. The results of the standing training should be briefly summarized in the results to provide the reader with context. 

Thank you for your suggestions. The result about standing balance training have been summarized in the introduction section of manuscript as suggested. In addition, the training protocol has been described in more detail in the supplementary material. 

Reviewer #3: This manuscript describes a study investigating the effect of balance training on neuromuscular and stability control in older adults during normal and narrow base walking. The findings indicated no training effects on these parameters. As expected, the base of support constraint resulted in walking adaptations in both neuromuscular and stability parameters. Although the manuscript is well organized, some aspects could be improved to facilitate readability to those not familiarized with some parameters analyzed in this study. The discussion could also be improved to explain the absence of effect of the balance training.

1- The abstract would benefit from the use of plain language, particularly when describing the results. The use of jargon (particularly FWHM and CoA) makes it difficult to follow and understand the main findings.

We thank the reviewer for this comment. We have updated the manuscript to incorporate the definition in plain language.

2- The first two sentences of the introduction suggest a logical interconnection that is not the case. The fact that people often fall during walking does not imply that a "balance training should transfer to gait and improve gait stability".

Thank you for the comment. The introduction has been modified as follows:

“Falls in older adults mostly occur during walking [1,2]. Thus, if skills acquired by balance training programs do not transfer to improvements in gait stability, they are unlikely to decrease the number of falls.” Lines 37-39.

3- How was the sample size determined for this study? Was there any sample size calculation?

Thank you for the comment. More details have been added to the manuscript regarding the sample size as follows: 

“The required sample size was estimated at twenty-two based on power analysis for an F test of a repeated measure, assuming an effect size of 0.44 and correlation among repeated measures of 0.6 (β = 0.8, G * power 3.1.9.2, Düsseldorf, Germany), comparable to similar studies [38,39].” Lines 132-135.

4- The description of the balance training protocol (page 6, lines 104-109) is very brief. It is essential to provide additional information regarding the duration of each session, the progression of the exercises throughout the sessions, and the organization of each session (warm-up, main activities, etc.).

We thank the reviewer for the suggestion. Additional information has been added as supplementary material. It now reads:

“Participants first trained for 30 minutes individually between pre and post1 measurement timepoint on a single day. Then, for three weeks with a frequency of three times per week, for 45 minutes per session, they trained in a group of 4 to 8 people. A training session consisted of blocks of 40-60 second exercises in which balance was challenged by different surface conditions, static conditions, perturbations, and dual tasks. Over the course of the training period, the difficulty level was increased by using more challenging exercises, challenging surface conditions (foam, balance boards), and perturbations. Progression criteria were based on the researcher’s observation during the training sessions; if participants were able to perform the task for 60 seconds, the difficulty level would be increased using different balance boards or by limiting the sensory inputs (Figure. S1.1.). Participants were encouraged to train at their individual balance ability level. To test the transfer of acquired skill to gait, none of the exercises included stepping, jumping, or locomotion. To maintain safety, exercises were carried out in groups of two under supervision of the researchers.”

5- When describing the FWHM and CoA (page 9, lines 186-192), include the meaning of these variables for those not familiarized with these variables.

Thank you for the comment. The following text has been added to the manuscript:

"The FWHM is defined as the number of data points above half of the maximum of the activation profile, after subtracting the minimum activation [56]. In addition, we evaluated the CoA (indicating the center of the distribution of activation timing within a gait cycle) per stride, defined as the angle of the vector that points to the center of mass in the activation profile transformed to polar coordinates [57]. The FWHM metric reflects the duration of activation but is naïve of timing of activation. The CoA metric reflects the timing of activation but is naïve of duration of activation.” Lines 263-269.

6- For the beam walking task, did the participants step off the beam at any moment? What was the maximum foot placement error accepted when the participant eventually did not step with the entire foot over the beam?

Thank you for the comment. We did not dichotomize success or failure in the beam walking trial. Participants stepped outside the beam sometimes, but this did not lead to task failure, as the beam was a virtual beam. Instead, we measured where participants placed their feet relative to the target (beam) pre and post training. No participant had a near-fall, nor did anyone have to stop the trial. The foot placement error was determined as the distance between the outer edge of the foot and the edge of the beam. We measured this difference for every step, with a foot placement error indicating that a part or the whole foot was placed outside the beam. When the foot was completely inside the beam, the error was zero. Based on a comment of the first reviewer, we have now added two new measures to our gait evaluation: step width and step width variability. After training we found decreased step width in narrow-base walking and decreased step width variability in both normal and narrow-base walking. 

7- Since the beam width (12 cm) is remarkably close to the average foot width, how foot placement error could be affected by balance training? Wouldn't you have a floor effect for this variable?

Thank you for the comment. It should be noted, when interpreting the 1.5 cm mean step error, that the width of the beam is 12 cm whilst average foot width is about 10 cm (Jurca, Zabkar, Dzeroski, 2019). This means that participants, on average, had a 2 cm margin in foot placement that was not seen as error. So, the task did not require perfect control. Additionally, the percentages of steps inside the beam were 27.6% (SE 3.9%) pre-training and 33.3% (SE 4.4%) post-training. Although these differences were not significant (p=0.11), we do believe this suggests that there is ample room for improvement. The best recorded performance was 67% of steps within the beam, which demonstrated to us that scores much higher than the mean could be obtained, and that even these best performances were not close to the end of the scale. 

The % of the steps within the beam has been added to the manuscript as below: 

“Also, the percentage of the steps within the beam did not change significantly with Training (F2,40= 2.934, p = 0.065; Fig 3.b).” Lines 293-294.

However, as we mentioned in previous response, we have added two new measures to our gait evaluation which shows different result. We found decreased step width in narrow-base and decreased step width variability in both conditions after training

8- Page 13 (lines 286-287): what do you mean by "robustness of gait"?

Thank you for the comment. Robustness of gait refers to the ability to prevent falls in the presence of perturbations and managing to regain balance quickly. However, this part is removed from the discussion. 

9- The discussion about the absence of effect of the balance training in all investigated variables was incredibly brief (page 14, lines 313-317). Most of the discussion is focused on the effects of the base of support manipulation. The discussion regarding the balance training needs to improve to explain the absence of an effect. Only arguing that the duration of the training protocol was short is not enough. Why did you choose a 3-week intervention if studies are showing that more time is needed to observe gait improvements? Were the variables not sensitive enough to capture gains due to balance training?

Thank you for your suggestions. Addition of the step width outcome measures, suggested by the reviewer, have changed interpretation of the effect of balance training. The discussion has been updated and restructured into the following sections general findings, effect of training, effect of narrow base, limitations, and conclusions. The section ‘Transfer of training effects’ in the discussion now has a more extensive discussion on the effect of training on the outcome measures.

The measures were sensitive enough to show the modulation between narrow-base and normal walking, therefore, the lack of sensitivity of the measures is unlikely. 

10- I recommend including a paragraph with the limitations of this study.

Thank you for your suggestion. The discussion has been updated and the limitations of the study are addressed in lines 508-525.

6. PLOS authors have the option to publish the peer review history of their article (what does this mean?). If published, this will include your full peer review and any attached files.

Yes

 1. Hu MH, Woollacott MH. Multisensory training of standing balance in older adults: II. Kinematic and electromyographic postural responses. Journals Gerontol 1994;49:.

2. Bohm S, Mandla-Liebsch M, Mersmann F, Arampatzis A. Exercise of Dynamic Stability in the Presence of Perturbations Elicit Fast Improvements of Simulated Fall Recovery and Strength in Older Adults: A Randomized Controlled Trial. Front Sport Act Living 2020;2:1–10.

3. Sawers A, Allen JL, Ting LH. Long-term training modifies the modular structure and organization of walking balance control. J Neurophysiol 2015;114:3359–3373.

4. Neptune RR, McGowan CP. Muscle contributions to frontal plane angular momentum during walking. J Biomech 2016;49:2975–2981.

5. Vieira TMM, Minetto MA, Hodson-Tole EF, Botter A. How much does the human medial gastrocnemius muscle contribute to ankle torques outside the sagittal plane? Hum Mov Sci 2013;32:753–767.

6. Sozzi S, Honeine JL, Do MC, Schieppati M. Leg muscle activity during tandem stance and the control of body balance in the frontal plane. Clin Neurophysiol 2013;124:1175–1186.

7. Hien TD, Tuan D Van, At P Van, Son LH. Novel algorithm for non-negative matrix factorization. New Math Nat Comput 2015;11:121–133.

8. Rubenstein LZ, Josephson KR, Robbins AS. Falls in the nursing home. Ann Intern Med 1994;121:442–451.

9. Berg WP, Alessio HM, Mills EM, Tong C. Circumstances and consequences of falls in independent community-dwelling older adults. Age Ageing 1997;26:261–268.

10. Sawers A, Allen JL, Ting LH. Long-term training modifies the modular structure and organization of walking balance control. J Neurophysiol 2015;114:3359–3373.

11. Chvatal SA, Ting LH. Common muscle synergies for balance and walking. Front Comput Neurosci 2013;7:1–14.

12. Bekius A, Bach MM, van der Krogt MM, de Vries R, Buizer AI, Dominici N. Muscle Synergies During Walking in Children With Cerebral Palsy: A Systematic Review. Front Physiol 2020;11:.

13. Chvatal SA, Torres-Oviedo G, Safavynia SA, Ting LH. Common muscle synergies for control of center of mass and force in nonstepping and stepping postural behaviors. J Neurophysiol 2011;106:999–1015.

---

## [Decision Letter · Decision Letter 1]

29 Mar 2022

PONE-D-20-30116R1Improvement in gait stability in older adults after ten sessions of standing balance trainingPLOS ONE

Dear Dr. van Dieën,

Thank you for submitting your manuscript to PLOS ONE. After careful consideration, we feel that it has merit but does not fully meet PLOS ONE’s publication criteria as it currently stands. Therefore, we invite you to submit a revised version of the manuscript that addresses the points raised during the review process.

 All three reviewers have found merit with your manuscript, however, there are still some suggested changes. Please pay particular attention to Reviewer 1 who has additional comments that need to be addressed within the manuscript (e.g. great justification in places).

We look forward to receiving your revised manuscript.

Kind regards,

Jeremy P Loenneke

Academic Editor

PLOS ONE

Journal Requirements:

Reviewers' comments:

Reviewer's Responses to Questions

**Comments to the Author**

1. If the authors have adequately addressed your comments raised in a previous round of review and you feel that this manuscript is now acceptable for publication, you may indicate that here to bypass the “Comments to the Author” section, enter your conflict of interest statement in the “Confidential to Editor” section, and submit your "Accept" recommendation.

Reviewer #1: (No Response)

Reviewer #2: All comments have been addressed

Reviewer #3: All comments have been addressed

2. Is the manuscript technically sound, and do the data support the conclusions?

Reviewer #1: Partly

Reviewer #2: Yes

Reviewer #3: Yes

3. Has the statistical analysis been performed appropriately and rigorously? 

Reviewer #1: Yes

Reviewer #2: Yes

Reviewer #3: Yes

4. Have the authors made all data underlying the findings in their manuscript fully available?

Reviewer #1: Yes

Reviewer #2: No

Reviewer #3: Yes

5. Is the manuscript presented in an intelligible fashion and written in standard English?

Reviewer #1: Yes

Reviewer #2: Yes

Reviewer #3: Yes

6. Review Comments to the Author

Reviewer #1: The changes based on the previous review have strengthened the manuscript. However, there are a few issues to be addressed to increase clarity and readability.

Major issues:

1. Justification for step width and step width variability based on stability/control of gait. While we (review team) suggested the authors examine this measure, the authors need to provide strong justifications for why these variables were included and add appropriate hypotheses. Line 107 and lines 122-126 are redundant (“The two latter variables were added to the analysis upon a reviewer’s suggestion” line 107). I appreciate that the authors clearly and explicitly state that SW and SWvar were not initially included. I recommend removing the statement from line 107 and keeping lines 122-126. However, you should not rely only on the suggestion of a reviewer, there must be a rationale beyond suggestion of a third party. Increase the justification for these measures based on stability/empirical data. You state earlier that SW and SWvar are associated with age and with older fallers (lines 65-67), expand on that.

2. Skill transfer. Justify the use of the terminology ‘skill transfer’. Step width and step width variability are, of course, not observed during standing, so explicitly state the skill that is transferred? What you observed is a change in gait parameters after standing balance training. This may or may not be ‘skill transfer’, so clarify what skill was transferred and the rationale/evidence that support this interpretation.

3. Adaptable and plastic. Line 376 - Authors used the set of terms – adaptable (modulated between conditions) and plastic (modified by training) – for the first time in the discussion. These terms increase clarity when describing/interpreting the different statistical outcomes. Thus, it would be helpful to include and define these terms in the introduction and perhaps build them into the purpose/aim/hypotheses. Similarly, line 384-386 seems to be related to the condition comparison effects, but not training effects. Be explicit. Please also explain if/how these terms relate to motor modules? We also suggest you use the terms in your headings.

Minor issues:

Line 37-38 – for clarity change “…skills acquired by balance training programs…” to “…skills acquired by standing balance training programs…” (clarification added as many balance training programs including gait activities)

Line 43 – similar comment as above – were the four references [4-8] only examining standing balance training, or balance training that went beyond standing tasks?

Line 48-56- This paragraph is not clear. What is an ‘altered modulation’ in the last sentence – this seems to be a change in gait that has changed (this may be related to the terms adaptable and plastic, and thus may increase clarity by adding the terms to the introduction)? Also, I don’t follow the logic – if skill transfer (from standing to gait) occurs, then you expect gait to be different between normal and narrow-base walking (NW and NBW, respectively)? Or do you expect a *greater* difference between NW and NBW after training than before training? If the latter is an accurate description of the expectation, why is that expected? Further, the following statement is unclear - “increased confidence after training may result in less adaptation to a challenging condition”. I would assume that increased confidence in a task would allow more adaptations. Also, confidence is included in this paragraph, but is only superficially addressed in one sentence, and is not included in the concluding statement of the paragraph.

Line 87 – ‘similarly widened activation profiles’ – similar to what?

Line 134 - Why did authors assume effect size of 0.44. Is it a partial eta squared value?

Lien 158 - specify perturbations (self-perturbation or externally applied, etc.)

Line 200- How did authors quantify leg dominance?

Line 428 - Authors compared motor modules of novice vs expert ballet dancers and specify that their training period was ten years. It would be helpful to relate how that fits in with the current results.

Line 451 - redundant text with the text in methods section

Editorial comments:

Line 246 - Underscores before the reference

Line 291 onwards - At multiple locations, “T” is capitalized for training in the middle of the sentence

Line 398- Be consistent in format for reporting mean +/- SE 27.6% (SE 3.9%) pre-training and 33.3% (SE 4.4%)

Reviewer #2: (No Response)

Reviewer #3: I want to thank the authors for their thoughtful revisions. The new results and the authors’ modifications improved the manuscript considerably. I have just a minor comment. In three parts of the manuscript (two in the Introduction and one in the Discussion), the authors commented on one of the reviewer’s suggestions of adding step width and step width variability variables to argue that these analyses were not pre-planned. You could mention this only in the Discussion (p. 19, lines 451-453). I do not see the reason to keep repeating this information in different parts of the manuscript.

7. PLOS authors have the option to publish the peer review history of their article (what does this mean?). If published, this will include your full peer review and any attached files.

Reviewer #1: No

Reviewer #2: No

Reviewer #3: No

---

## [Author Response · Author response to Decision Letter 1]

3 May 2022

We would like to thank the reviewers for their valuable comments. Without their comments the article would have not reached this level of quality. Below there are point-by-point answers to reviewers’ comments. 

Reviewer #1: The changes based on the previous review have strengthened the manuscript. However, there are a few issues to be addressed to increase clarity and readability.

Major issues:

1. Justification for step width and step width variability based on stability/control of gait. While we (review team) suggested the authors examine this measure, the authors need to provide strong justifications for why these variables were included and add appropriate hypotheses. 

Line 107 and lines 122-126 are redundant (“The two latter variables were added to the analysis upon a reviewer’s suggestion” line 107). I appreciate that the authors clearly and explicitly state that SW and SWvar were not initially included. I recommend removing the statement from line 107 and keeping lines 122-126. 

However, you should not rely only on the suggestion of a reviewer, there must be a rationale beyond suggestion of a third party. Increase the justification for these measures based on stability/empirical data. You state earlier that SW and SWvar are associated with age and with older fallers (lines 65-67), expand on that.

Thank you for the suggestion. Line 107 has been removed. The texts below have been added to the introduction:

(Lines 67-74): Mediolateral gait stability is thought to be actively controlled by adjusting foot placement (Bruijn & Van Dieën, 2018), as centre of mass kinematics during the preceding swing phase strongly correlate with foot position in the next stance phase (Hurt et al., 2010; Wang & Srinivasan, 2014). The strong coupling between centre of mass kinematics and foot placement is found to decrease in conditions in which gait is stabilized, such as in external lateral stabilization, increasing confidence that lateral gait stability is indeed controlled by foot placement adjustment (Mahaki et al., 2019). This decoupling coincides with a decrease in step width and step width variability (Donelan et al., 2004). Moreover, increased step width and step width variability have been found in older compared to young adults and in fallers compared to non-fallers (Callisaya et al., 2010; Nordin et al., 2010; Skiadopoulos et al., 2020). 

(Lines 76-78): Consequently, step width and step width variability are considered to be proxies of the quality of active mediolateral control of gait stability using foot placement adjustments and may therefore be useful to evaluate the effect of balance training.

(Lines 135-146): Even though step width and step width variability seem to be sensitive to quality of balance control in gait, step width measures are not necessarily affected by postural balance training. Postural balance is controlled using torques around the ankle and hip of the standing leg (Nashner, 1985; Runge et al., 1999), but does not include stepping (i.e. foot placement). However, as in postural balance, the stance leg is used to regulate stabilizing torques in the stance phase gait (Reimann et al., 2018). This stance leg control co-determines the control of foot placement (van Leeuwen et al., 2021) and therefore reduced step width and step width variability are likely results of improved control by the stance leg. These predictions on step width and step width variability were conceived, based on reviewer comments, after the data were processed and analyzed. These predictions are thus explorative and aim to drive future research (Kerr, 1998; Rowbottom & Alexander, 2012) and we ask the reader to consider the limitations of the evidence provided by these variables. 

2. Skill transfer. Justify the use of the terminology ‘skill transfer’. Step width and step width variability are, of course, not observed during standing, so explicitly state the skill that is transferred? What you observed is a change in gait parameters after standing balance training. This may or may not be ‘skill transfer’, so clarify what skill was transferred and the rationale/evidence that support this interpretation.

We thank the reviewer for the comment. We agree that strictly speaking we do not provide evidence for transfer of acquired skill, while we do provide evidence for transfer of an effect of training. We now use the more neutral term transfer, omitting skill. There are changes throughout the manuscript in Key words, introduction, and discussion.

Lines 37-39: Thus, if effects of standing balance training programs do not transfer to improvements in gait stability, they are unlikely to decrease the number of falls.

Lines 44-46: Consequently, the existence of transfer from standing balance training to gait, as well as the mechanisms underlying such a transfer, if present, are insufficiently clear.

Lines 109-113: In the current study, we aimed to investigate the transfer of effects of standing balance training to normal and narrow-base walking in older adults, as well as the adaptation of older adults to narrow-base walking. 

Lines 146-148: While this result is in line with our primary hypothesis and indicates transfer of effects of standing balance training to gait, in a strict sense it cannot be considered a planned analysis.

Line 398: We studied whether effects of standing balance training transferred to gait in older adults.

Lines 403-404: We also expected transfer of training effects to be most pronounced in the narrow-base condition.

Lines 415-416: In this study, after ten sessions we found transfer from standing balance training to gait.

Lines 474-475: (to start with stationary exercises and transfer the training effects to walking)

3. Adaptable and plastic. Line 376 - Authors used the set of terms – adaptable (modulated between conditions) and plastic (modified by training) – for the first time in the discussion. These terms increase clarity when describing/interpreting the different statistical outcomes. Thus, it would be helpful to include and define these terms in the introduction and perhaps build them into the purpose/aim/hypotheses. Similarly, line 384-386 seems to be related to the condition comparison effects, but not training effects. Be explicit. Please also explain if/how these terms relate to motor modules? We also suggest you use the terms in your headings.

Thank you for the comment. We agree these two terms should have been introduced earlier in the introduction. We now introduce adaptable and plastic in the introduction. 

Lines 111- 113: The modulation of balance control between two conditions aimed to test adaptability of balance control to environmental constraints and effects of training were studied to analyse the plasticity of balance control. 

Lines 48-50: Fall prevention training programs aim to improve balance control employing plastsicity of the neuromuscular system . To prevent falls, one needs to be able to adapt gait when facing environmental challenges, such as when forced to walk with a narrow step width.

Minor issues:

Line 37-38 – for clarity change “…skills acquired by balance training programs…” to “…skills acquired by standing balance training programs…” (clarification added as many balance training programs including gait activities)

Thank you for the comment. Skills acquired by balance training programs is now changed to effects of standing balance training programs as suggested.

Line 43 – similar comment as above – were the four references [4-8] only examining standing balance training, or balance training that went beyond standing tasks?

Studies 4, 5, and 8 refer to standing balance training. Study 6 used gait and 7 a combination of gait and standing and sitting, lying on the floor, etc. The latter two references have thus been removed.

The text now reads as (Lines 41-43): On the other hand, transfer to gait stability from solely standing balance training [4-6] is suggested by improved clinical balance scores, gait parameters, and performance on the timed up and go, and other tests.

Line 48-56- This paragraph is not clear. What is an ‘altered modulation’ in the last sentence – this seems to be a change in gait that has changed (this may be related to the terms adaptable and plastic, and thus may increase clarity by adding the terms to the introduction)? Also, I don’t follow the logic – if skill transfer (from standing to gait) occurs, then you expect gait to be different between normal and narrow-base walking (NW and NBW, respectively)? Or do you expect a *greater* difference between NW and NBW after training than before training? If the latter is an accurate description of the expectation, why is that expected? Further, the following statement is unclear - “increased confidence after training may result in less adaptation to a challenging condition”. I would assume that increased confidence in a task would allow more adaptations. Also, confidence is included in this paragraph, but is only superficially addressed in one sentence, and is not included in the concluding statement of the paragraph.

Thank you for the comments. Indeed, we expect a change in the difference between NW and NBW after training, that is an interaction between Training and Condition. However, we were unsure about the direction of this effect. We have rephrased this section to (Lines: 49-58): 

“To prevent falls, one needs to be able to adapt gait when facing environmental challenges, such as when forced to walk with a narrow step width. Older adults show more pronounced adaptations to narrow-base walking than young adults [11], possibly because they are more cautious in the presence of balance threats [12]. Therefore, an interaction between training and stabilizing demands may be expected. On the one hand, increased confidence after training may result in less adaptation to a challenging condition. On the other hand, balance training may enhance the ability to adapt to challenging conditions. Therefore, if transfer of standing balance training to gait occurs, an altered modulation of gait between normal and narrow-base walking might be expected after training, but the direction of change is unpredictable.”

We expected that training might increase confidence. We did not elaborate on the confidence in the main manuscript, since it was not the focus of the research. However, we provided as short introduction to the topic and the results in the supplementary material for curious readers to test our speculation (copied below). 

Several studies showed a strong correlation between concern of falling and balance performance (Thiamwong & Suwanno, 2014; Young & Mark Williams, 2015). It has been shown that poor balance performance is mediated by changes in the allocation of attention in the presence of concern of falling [(Young & Mark Williams, 2015)]. Concern of falling is reduced after training in older adults, which is associated with improved balance performance (Kumar et al., 2016; Thiamwong & Suwanno, 2014). To assess concern of falling, we used the Falls Efficacy Scale International (FES-I) questionnaire at pre, post2, and retention time-points (Kempen et al., 2007). FES-I outcomes are on a scale of 16 to 64, with 16 indicating minimum concern of falling and 64 severe concern of falling.

A repeated measures ANOVA indicated that concern of falling was affected by balance training (F2,42= 4.37, P = 0.039; Figure S2.1). Post-hoc analysis showed that concern of falling was not significantly changed immediately after the training program, but was decreased at retention (t = 2.16, p = 0.072; t = 2.82, p = 0.022, respectively), implying that weeks after training participants felt more confident about their balance ability.

Line 87 – ‘similarly widened activation profiles’ – similar to what?

Thank you for the comment. The word “similarly” has been removed.

Line 134 - Why did authors assume effect size of 0.44. Is it a partial eta squared value?

We chose this effect size from the meta-analysis article, DOI 10.1007/s40279-015-0375-y, which expressed effects size using Cohen’s f. This has now been mentioned in the text as:

Lines 154-158: The required sample size was estimated at twenty-two based on power analysis for an F test of a repeated measures ANOVA, assuming a Cohen’s f of 0.44 (based on meta-analysis of the effect of standing balance training on steady-state balance (Muehlbauer et al., 2015)) and a correlation among repeated measures of 0.6 (β = 0.8, G * power 3.1.9.2, Düsseldorf, Germany), comparable to similar studies (Bisson et al., 2007; Nagy et al., 2007).

Line 158 - specify perturbations (self-perturbation or externally applied, etc.)

We have further specified the perturbations, and the text now reads (Lines 181-182): self-perturbations and external perturbations while catching a ball in a dual tasking exercise.

Line 200- How did authors quantify leg dominance?

We have added this to the methods section as (Lines 225-227): The preferred stance leg was reported by the participant prior to the experiment and confirmed by the experimenter by asking the participant to kick an imaginary soccer ball. The supporting leg was considered the preferred stance leg (Alizadehsaravi et al., 2022).

Line 428 - Authors compared motor modules of novice vs expert ballet dancers and specify that their training period was ten years. It would be helpful to relate how that fits in with the current results.

The mentioned study was a cross-sectional study, in which the experts had a minimum of 10 years training, including all sorts of training in ballet, which could vary from standing balance to gait and more, but with balance control as a core element. The aim of mentioning this study was to show that a long-term training may cause reorganization and permanent changes.

The text (Lines 447-455) now reads as: Even though we found evidence of transfer, gait synergies were not affected by training. Synergy metrics may be insensitive, because subtle variations in the activation of some muscles, for example through improved feedback control, underlie improved balance. Alternatively, changes may have occurred in muscles that we did not include in our measurements. Young adults with years of experience in balance training in ballet showed different motor modules during narrow-base walking compared to novice ballet dancers (Chvatal & Ting, 2013), indicating that long-term effects of training may be apparent in muscle synergies. Note, however, that these ballet dancers’ training experience most certainly included gait-related tasks (Chvatal & Ting, 2013). In addition, the dancers had trained for at least ten years. 

Line 451- redundant text with the text in methods section

We removed the redundant text and kept the one in the introduction.

Editorial comments: Line 246 - Underscores before the reference

The underscores have been removed, thank you for noticing that. 

Line 291 onwards - At multiple locations, “T” is capitalized for training in the middle of the sentence

Thank you for the comment. We capitalized the T intentionally where we refer to Training as a factor in the ANOVA. 

Line 398- Be consistent in format for reporting mean +/- SE 27.6% (SE 3.9%) pre-training and 33.3% (SE 4.4%)

Thanks. We reported the SD now. 

Reviewer #2: (No Response) 

Reviewer #3: I want to thank the authors for their thoughtful revisions. The new results and the authors’ modifications improved the manuscript considerably. I have just a minor comment. In three parts of the manuscript (two in the Introduction and one in the Discussion), the authors commented on one of the reviewer’s suggestions of adding step width and step width variability variables to argue that these analyses were not pre-planned. You could mention this only in the Discussion (p. 19, lines 451-453). I do not see the reason to keep repeating this information in different parts of the manuscript.

Thank you for the compliment and the time you devoted to review our paper. We have now removed the first statement and acknowledged the changes at the end of the introduction as reviewer #1 suggested and the discussion as you suggested.

References:

Alizadehsaravi, L., Koster, R. A. J., Muijres, W., Maas, H., Bruijn, S. M., & van Dieën, J. H. (2022). The underlying mechanisms of improved balance after one and ten sessions of balance training in older adults. Human Movement Science, 81(December 2021), 1–14. https://doi.org/10.1016/j.humov.2021.102910

Bisson, E., Contant, B., Sveistrup, H., & Lajoie, Y. (2007). Functional balance and dual-task reaction times in older adults are improved by virtual reality and biofeedback training. Cyberpsychology and Behavior, 10(1), 16–23. https://doi.org/10.1089/cpb.2006.9997

Bruijn, S. M., & Van Dieën, J. H. (2018). Control of human gait stability through foot placement. Journal of the Royal Society Interface, 15(143). https://doi.org/10.1098/rsif.2017.0816

Callisaya, M. L., Blizzard, L., Schmidt, M. D., McGinley, J. L., & Srikanth, V. K. (2010). Ageing and gait variability-a population-based study of older people. Age and Ageing, 39(2), 191–197. https://doi.org/10.1093/ageing/afp250

Chvatal, S. A., & Ting, L. H. (2013). Common muscle synergies for balance and walking. Frontiers in Computational Neuroscience, 7(May), 1–14. https://doi.org/10.3389/fncom.2013.00048

Donelan, J. M., Shipman, D. W., Kram, R., & Kuo, A. D. (2004). Mechanical and metabolic requirements for active lateral stabilization in human walking. Journal of Biomechanics, 37(6), 827–835. https://doi.org/10.1016/j.jbiomech.2003.06.002

Hurt, C. P., Rosenblatt, N., Crenshaw, J. R., & Grabiner, M. D. (2010). Variation in trunk kinematics influences variation in step width during treadmill walking by older and younger adults. Gait and Posture, 31(4), 461–464. https://doi.org/10.1016/j.gaitpost.2010.02.001

Kempen, G. I. J. M., Zijlstra, G. A. R., & van Haastregt, J. C. M. (2007). [The assessment of fear of falling with the Falls Efficacy Scale-International (FES-I). Development and psychometric properties in Dutch elderly]. Tijdschrift voor gerontologie en geriatrie, 38(4), 204–212.

Kerr, N. L. (1998). HARKing: Hypothesizing after the results are known. Personality and Social Psychology Review, 2(3), 196–217. https://doi.org/10.1207/s15327957pspr0203_4

Kumar, A., Delbaere, K., Zijlstra, G. A. R., Carpenter, H., Iliffe, S., Masud, T., Skelton, D., Morris, R., & Kendrick, D. (2016). Exercise for reducing fear of falling in older people living in the community: Cochrane systematic review and Meta-Analysis. Age and Ageing, 45(3), 345–352. https://doi.org/10.1093/ageing/afw036

Mahaki, M., Bruijn, S. M., & Van Dieën, J. H. (2019). The effect of external lateral stabilization on the use of foot placement to control mediolateral stability in walking and running. PeerJ, 2019(10). https://doi.org/10.7717/peerj.7939

Muehlbauer, T., Gollhofer, A., Lesinski, M., Hortoba, T., & Granacher, U. (2015). Effects of Balance Training on Balance Performance in Healthy Older Adults : A Systematic Review and Meta-analysis. Sports Med, 45(45), 1721–1738. https://doi.org/10.1007/s40279-015-0375-y

Nagy, E., Feher-Kiss, A., Barnai, M., Domján-Preszner, A., Angyan, L., & Horvath, G. (2007). Postural control in elderly subjects participating in balance training. European Journal of Applied Physiology, 100(1), 97–104. https://doi.org/10.1007/s00421-007-0407-x

Nashner, L. M. (1985). The organization of human postural movements : A formal basis and experimental synthesis. 8, 135–172. https://doi.org/10.1017/S0140525X00020008

Nordin, E., Moe-Nilssen, R., Ramnemark, A., & Lundin-Olsson, L. (2010). Changes in step-width during dual-task walking predicts falls. Gait & Posture, 32(1), 92–97. https://doi.org/https://doi.org/10.1016/j.gaitpost.2010.03.012

Reimann, H., Fettrow, T., & Jeka, J. J. (2018). Strategies for the Control of Balance During Locomotion. Kinesiology Review, 7(1), 18–25. https://doi.org/10.1123/kr.2017-0053

Rowbottom, D. P., & Alexander, R. M. N. (2012). The role of hypotheses in biomechanical research. Science in Context, 25(2), 247–262. https://doi.org/10.1017/S0269889712000051

Runge, C. F., Shupert, C. L., Horak, F. B., & Zajac, F. E. (1999). Ankle and hip postural strategies defined by joint torques. Gait and Posture, 10(2), 161–170. https://doi.org/10.1016/S0966-6362(99)00032-6

Skiadopoulos, A., Moore, E. E., Sayles, H. R., Schmid, K. K., & Stergiou, N. (2020). Step width variability as a discriminator of age-related gait changes. Journal of NeuroEngineering and Rehabilitation, 17(1), 1–13. https://doi.org/10.1186/s12984-020-00671-9

Thiamwong, L., & Suwanno, J. (2014). Effects of simple balance training on balance performance and fear of falling in rural older adults. International Journal of Gerontology, 8(3), 143–146. https://doi.org/10.1016/j.ijge.2013.08.011

van Leeuwen, A. M., van Dieën, J. H., Daffertshofer, A., & Bruijn, S. M. (2021). Ankle muscles drive mediolateral center of pressure control to ensure stable steady state gait. Scientific Reports, 11(1), 1–14. https://doi.org/10.1038/s41598-021-00463-8

Wang, Y., & Srinivasan, M. (2014). Stepping in the direction of the fall: The next foot placement can be predicted from current upper body state in steady-state walking. Biology Letters, 10(9). https://doi.org/10.1098/rsbl.2014.0405

Young, W. R., & Mark Williams, A. (2015). How fear of falling can increase fall-risk in older adults: Applying psychological theory to practical observations. Gait and Posture, 41(1), 7–12. https://doi.org/10.1016/j.gaitpost.2014.09.006

---

## [Decision Letter · Decision Letter 2]

29 Jun 2022

Improvement in gait stability in older adults after ten sessions of standing balance training

PONE-D-20-30116R2

Dear Dr. van Dieën,

We’re pleased to inform you that your manuscript has been judged scientifically suitable for publication and will be formally accepted for publication once it meets all outstanding technical requirements.

Kind regards,

Jeremy P Loenneke

Academic Editor

PLOS ONE

Additional Editor Comments (optional):

Reviewers' comments:

Reviewer's Responses to Questions

**Comments to the Author**

1. If the authors have adequately addressed your comments raised in a previous round of review and you feel that this manuscript is now acceptable for publication, you may indicate that here to bypass the “Comments to the Author” section, enter your conflict of interest statement in the “Confidential to Editor” section, and submit your "Accept" recommendation.

Reviewer #1: (No Response)

Reviewer #2: All comments have been addressed

Reviewer #3: All comments have been addressed

2. Is the manuscript technically sound, and do the data support the conclusions?

Reviewer #1: Yes

Reviewer #2: Yes

Reviewer #3: Yes

3. Has the statistical analysis been performed appropriately and rigorously? 

Reviewer #1: Yes

Reviewer #2: Yes

Reviewer #3: Yes

4. Have the authors made all data underlying the findings in their manuscript fully available?

Reviewer #1: Yes

Reviewer #2: Yes

Reviewer #3: Yes

5. Is the manuscript presented in an intelligible fashion and written in standard English?

Reviewer #1: Yes

Reviewer #2: Yes

Reviewer #3: Yes

6. Review Comments to the Author

Reviewer #1: The authors have addressed all of my concerns. This is nice piece of work, and will stimulate other research.

Reviewer #2: (No Response)

Reviewer #3: I have no further comments. The authors addressed all concerns raised in the previous rounds. Congratulations on the manuscript.

7. PLOS authors have the option to publish the peer review history of their article (what does this mean?). If published, this will include your full peer review and any attached files.

Reviewer #1: No

Reviewer #2: No

Reviewer #3: No

---

## [Editor Report · Acceptance letter]

5 Jul 2022

PONE-D-20-30116R2 

Improvement in gait stability in older adults after ten sessions of standing balance training. 

Dear Dr. van Dieën:

I'm pleased to inform you that your manuscript has been deemed suitable for publication in PLOS ONE. Congratulations! Your manuscript is now with our production department. 

Kind regards, 

on behalf of

Dr. Jeremy P Loenneke 

Academic Editor

PLOS ONE